# Mediators of the association between psychological distress and mortality in people diagnosed with cancer

Natalie Ella Miller [1] ✉, Jaana Pentti[2], Andrew Steptoe [1], Mika Kivimaki [2,3], Phillippa Lally [4], Philipp Frank[3,5] & Abigail Fisher[1,5]

The biological and behavioural mechanisms linking psychological distress to excess mortality in people living with and beyond cancer (LWBC) remain poorly understood. In this multi-cohort study, we examine the associations between psychological distress and both cancer-specific and all-cause mortality in individuals living with and beyond cancer (LWBC), and assess the potential mediating roles of inflammation and health behaviours. The primary analysis includes 13,349 adults from the UK Biobank (2006–2021), with replication in 5739 participants from the Finnish Public Sector study (2000–2018). Psychological distress is associated with an increased risk of all-cause mortality (pooled adjusted risk ratio (RR), 1.54; 95% confidence interval (CI), 1.37–1.74) and cancer mortality (pooled RR, 1.57; 95% CI, 1.37–1.79). Systemic inflammation explains up to 18.6% of these associations, whereas diet, alcohol consumption, and body mass index do not mediate these relationships. Evidence for mediation by other health behaviours is inconsistent. While adjustment for cancer stage attenuates the distress-mortality link by up to 25%, psychological distress remains a robust predictor of both all-cause and cancer mortality. These findings suggest that psychological distress is an independent predictor of mortality risk in people with LWBC, partially attributable to elevated levels of systemic inflammation.

Advances in cancer detection and treatments have led to a growing number of people living with and beyond cancer (LWBC)[1]. Estimates suggest that by 2040, the number of people with LWBC will rise to over 5 million worldwide, compared with 3.5 million in 2025[1]. Despite improvements in survival rates, a cancer diagnosis often results in significant levels of psychological distress[2,3]. A previous meta-analysis of 70 studies found that, in adults LWBC, the prevalence of minor depression was 19%, major depression 15%, and anxiety 10%[4]. These estimates varied based on factors such as age, sex, cancer type, stage, and prognosis[5–7]. Additionally, a case-control study of 4020 adults with

cancer and 5018 control participants without cancer reported that the odds of depression were more than five times greater in people with cancer than in the general population[8].

Psychological distress is associated with a range of adverse outcomes in people with LWBC, including poorer quality of life[9], lower treatment adherence[10], and ultimately, poorer survival[11,12]. A recent meta-analysis found that depression and anxiety were associated with an 18% increased risk of cancer mortality (10 studies) and a 23% increased risk of all-cause mortality (15 studies) in people diagnosed with cancer[12]. However, the mechanisms through which psychological

[1]Department of Behavioural Science and Health, Institute of Epidemiology and Health Care, University College London, London, UK. [2]Clinicum, Faculty of Medicine, University of Helsinki, The Finnish Institute of Occupational Health, Helsinki, Finland. [3]UCL Brain Sciences, University College London, London, UK. [4]Department of Psychological Sciences, University of Surrey, Surrey, UK. [5]These authors contributed equally: Philipp Frank, Abigail Fisher. ✉e-mail: Natalie.miller.20@ucl.ac.uk

**Table 1 | Hazard ratios for the association between psychological distress and all-cause mortality after adjustment for each mediator**

| | UK Biobank | | FPS | |
|---|---|---|---|---|
| Model 2 | Adjusted HRs (95% CI) | - | Adjusted HRs (95% CI) | - |
| | 1.43 (1.23–1.67) | - | 1.72 (1.44–2.07) | - |
| Model 2 + cancer stage | Adjusted HRs (95% CI) | - | Adjusted HRs (95% CI) | % of attenuation |
| | - | - | 1.56 (1.29–1.88) | 22.2 |
| Model 2 + mediator | Adjusted HRs (95% CI) | % of attenuation | Adjusted HRs (95% CI) | % of attenuation |
| Inflammation | 1.35 (1.15–1.57) | 18.6 | - | - |
| Fruit | 1.42 (1.22–1.66) | 2.3 | - | - |
| Vegetables | 1.43 (1.23–1.67) | 0.0 | - | - |
| Red meat | 1.43 (1.23–1.67) | 0.0 | - | - |
| Processed meat | 1.43 (1.23–1.67) | 0.0 | - | - |
| Physical activity | 1.40 (1.20–1.63) | 7.0 | 1.68 (1.40–2.02) | 5.6 |
| BMI | 1.44 (1.23–1.67) | 0.0 | 1.73 (1.44–2.08) | 0.0 |
| Alcohol | 1.42 (1.22–1.66) | 2.3 | 1.73 (1.44–2.08) | 0.0 |
| Smoking | 1.36 (1.17–1.59) | 16.3 | 1.71 (1.43–2.06) | 1.4 |
| Sleep quality | 1.42 (1.21–1.65) | 2.3 | 1.70 (1.40–2.06) | 2.8 |
| Sleep duration | 1.39 (1.19–1.62) | 9.3 | 1.75 (1.45–2.11) | 0.0 |

Cox proportional hazards models (two-tailed).
CI = 95% confidence interval.
All models adjusted for age, sex, ethnicity, education, number of comorbidities, age at cancer diagnosis, time between cancer diagnosis and depressive symptoms assessment, and antidepressant medication (yes/no).

distress affects survival in people LWBC remain poorly understood. One hypothesis is that elevated inflammation plays a role, as inflammation has been linked to depression across various populations[13,14] and has also been associated with several stages of cancer progression, including cellular transformation, promotion, survival, proliferation, invasion, angiogenesis, and metastasis[15–17]. To date, only one previous study has assessed the mediating role of inflammation in the association between depression and survival in people LWBC[18]. This study found that systemic inflammation fully mediated the association between depression and survival. This conclusion remains uncertain because the study was conducted among a small sample of 394 patients with head and neck cancer.

Other mechanisms through which psychological distress might impact survival in people LWBC include unhealthy behaviours such as poor diet, physical inactivity, smoking, high alcohol consumption, and inadequate sleep. These behaviours have been associated with an increased risk of depression and mortality in people LWBC[19–21]. However, no study has yet assessed the extent to which these behaviours mediate the relationship between distress and survival in this population. Understanding the mechanisms linking psychological distress to survival is crucial, as it could guide the development of interventions to improve survival among people with LWBC experiencing distress.

In this work, we use data from two large prospective cohort studies to examine the association of psychological distress with both cancer mortality and all-cause mortality in people LWBC, as well as the role of inflammation and health behaviours in these associations. We show that psychological distress is associated with an increased risk of all-cause and cancer mortality, partly due to high systemic inflammation. However, evidence for mediation by health behaviours is found to be inconsistent.

## Results
### Sample characteristics
Supplementary Dataset 1 shows the characteristics of the final analytical samples in both the primary and replication studies (see Supplementary Dataset 2 for a comparison between the imputed and observed data in UK Biobank). The UK Biobank sample included 13,349

participants (mean age = 60, SD = 7.1), of whom 6836 were female (51.2%) and 6513 were male (48.8%). The Finnish Public Sector study (FPS) sample included 5739 participants (mean age 57.5, SD 10.2), of whom 4752 were female (82.8%) and 987 were male (17.2%). Mean duration of follow-up was 11 years (SD 2.9) in UK Biobank and 6 years (SD 4.9) in FPS. During the follow-up period in the UK Biobank, 2421 (18.1%) participants died, including 1790 (13.4%) from cancer. In FPS, 505 (8.8%) died, of whom 440 (7.7%) died from cancer. A higher proportion of participants reported psychological distress in FPS (26%) compared with UK Biobank (6%).

### Association of psychological distress with all-cause mortality
In the UK Biobank, psychological distress was associated with a 1.43 times increased risk of all-cause mortality after adjustment for all covariates (95% CI, 1.23–1.67) (Supplementary Dataset 3). Additional adjustment for inflammation reduced the association by 18.6%, smoking 16.3%, sleep duration 9.3%, physical activity 7%, sleep quality 2.3%, alcohol consumption 2.3%, and fruit intake 2.3% (Table 1). In contrast, adjustment for vegetable, red meat, or processed meat intake, and body mass index (BMI), did not attenuate the association between psychological distress and all-cause mortality. In analyses stratified by cancer type, psychological distress was associated with an increased risk for all-cause mortality after adjustment for all covariates in non-melanoma skin cancer, breast cancer, and multiple myeloma.

In FPS, the results were largely consistent with those observed in the UK Biobank. Psychological distress was associated with a 1.72-fold increased risk of all-cause mortality after multivariable adjustment (95% CI: 1.44–2.07; Supplementary Dataset 3). Additional adjustment for cancer stage attenuated the association by 22%, but the relationship remained statistically significant (Table 1). Further adjustment for physical activity (added to the multivariable model before cancer stage was added) reduced the association by 5.6%, sleep quality by 2.8%, and by smoking by 1.4% (Table 1). Alcohol, BMI, and sleep duration did not attenuate the association. Psychological distress was also associated with an increased risk of all-cause mortality in individuals diagnosed with colorectal, female reproductive, and unspecified or other cancers.

To examine the combined mediating effect of biological and behavioural factors simultaneously, we applied the inverse odds ratio-

**Table 2 | Hazard ratios for the association between psychological distress and cancer mortality after adjustment for each mediator**

| | UK Biobank | | FPS | |
|---|---|---|---|---|
| Model 2 | Adjusted HRs (95% CI) | - | Adjusted HRs (95% CI) | - |
| | 1.45 (1.21–1.74) | - | 1.72 (1.41–2.10) | - |
| Model 2 + cancer stage | Adjusted HRs (95% CI) | - | Adjusted HRs (95% CI) | % of attenuation |
| | - | - | 1.54 (1.26–1.90) | 25.0 |
| Model 2 + mediator | Adjusted SHRs (95% CI) | % of attenuation | Adjusted SHRs (95% CI) | % of attenuation |
| Inflammation | 1.37 (1.14–1.64) | 17.8 | - | - |
| Fruit | 1.44 (1.20–1.72) | 2.2 | - | - |
| Vegetables | 1.45 (1.21–1.74) | 0.0 | - | - |
| Red meat | 1.45 (1.21–1.74) | 0.0 | - | - |
| Processed meat | 1.45 (1.21–1.74) | 0.0 | - | - |
| Physical activity | 1.42 (1.18–1.70) | 6.7 | 1.68 (1.38–2.05) | 5.6 |
| BMI | 1.45 (1.21–1.74) | 0.0 | 1.72 (1.41–2.11) | 0.0 |
| Alcohol | 1.44 (1.20–1.73) | 2.2 | 1.73 (1.41–2.11) | 0.0 |
| Smoking | 1.39 (1.16–1.67) | 13.3 | 1.72 (1.40–2.10) | 0.0 |
| Sleep quality | 1.44 (1.20–1.73) | 2.2 | 1.66 (1.35–2.04) | 8.3 |
| Sleep duration | 1.40 (1.20–1.69) | 11.1 | 1.75 (1.43–2.14) | 0.0 |

Competing risk regression (two-tailed).
CI = 95% confidence interval.
All models adjusted for age, sex, ethnicity, education, number of comorbidities, age at cancer diagnosis, time between cancer diagnosis and depressive symptoms assessment, and antidepressant medication (yes/no).

weighted method. In the UK Biobank, this approach indicated that approximately 29% (95% CI: −46.9, 104.8) of the association between psychological distress and all-cause mortality was mediated by the combined set of mediators. In FPS, the combined mediators accounted for 25.7% (95% CI: 1.7–49.7) of the association in the multivariable adjusted model (Model 2). After further adjustment for cancer stage (Model 3), the mediation estimate was similar (25.6%) but was no longer statistically significant (95% CI: −6.4, 57.5).

### Association of psychological distress with cancer mortality
In UK Biobank, psychological distress was associated with a 1.45-fold increased risk of cancer mortality after adjustment for all covariates (95% CI, 1.21-1.74) (Supplementary Dataset 4). Additional adjustment for inflammation reduced the association by 17.8%, smoking 13.3%, sleep duration 11.1%, physical activity 6.7%, sleep quality 2.2%, alcohol 2.2%, and fruit consumption by 2.2% (Table 2). Additional adjustment for vegetable, red meat, and processed meat intake, as well as BMI, did not attenuate the association between distress and cancer mortality. In analyses stratified by cancer type, distress was associated with an increased risk of cancer mortality after adjustment for all covariates in the following cancers: head and neck, breast, and multiple myeloma.

In FPS, distress was also associated with a 1.72-fold increased risk of cancer mortality after multivariable adjustment (95% CI, 1.41–2.10) (Supplementary Dataset 4). Additional adjustment for cancer stage attenuated the association by 25%, although the relationship remained statistically significant (Table 2). Further adjustment for sleep quality reduced the association by 8.3% and physical activity by 5.6%. Alcohol, smoking, BMI and sleep duration did not attenuate the association (Table 2). In cancer type-stratified analyses, distress was associated with a higher risk of cancer mortality in the following cancer types: colorectal, female reproductive, and unspecified or other.

To examine the combined mediating effect of biological and behavioural factors simultaneously, we applied the inverse odds ratio-weighted method. In the UK Biobank, this approach indicated that approximately 30% (95% CI: −55.7, 115.7) of the association between psychological distress and cancer mortality was mediated by the combined set of mediators. In FPS, the combined mediators accounted for 28.5% (95% CI: 3.1–53.9) of the association in the multivariable adjusted model (Model 2). After further adjustment for cancer stage (Model 3), the mediation estimate was similar (30%) but was no longer statistically significant (95% CI: −6.5, 66.4).

### Secondary analyses
In UK Biobank, psychological distress experienced 1-4 years **prior** to cancer diagnosis was associated with a 1.31 times increased risk of all-cause mortality in people LWBC after adjustment for all covariates (95%. CI: 1.13–1.51) (Table 3). Furthermore, psychological distress was associated with a 1.66-fold increased risk of all-cause mortality in people without cancer (95% CI: 1.57–1.76).

In FPS, distress was associated with 1.12 times increased risk of all-cause mortality in people without cancer (95% CI: 1.03–1.21) (Table 3). However, psychological distress experienced 1–4 years prior to cancer diagnosis was not associated with increased risk of all-cause mortality (HR = 1.02; 95% CI: 0.87–1.21).

### Meta-analyses
Pooling of the adjusted effect estimates from UK Biobank and FPS indicated that psychological distress was associated with a 1.54 times increased risk of all-cause mortality (95% CI, 1.37–1.74). When stratified by cancer type, psychological distress was associated with an increased risk of all-cause mortality in the following cancer types: colorectal (RR, 1.88; 95% CI, 1.22–2.90), non-melanoma skin (RR, 1.88; 95% CI, 1.27–2.78), breast (RR = 1.39; 95% CI, 1.11–1.74), female reproductive (RR = 1.61; 95% CI, 1.02–2.54), and unspecified or other (RR = 1.80; 95% CI, 1.17–2.76).

Psychological distress was also associated with a 1.57-fold increased risk of cancer mortality (95% CI, 1.37–1.79). When stratified by cancer type, distress was associated with an increased risk of cancer mortality in the following cancer types: head and neck (RR = 2.46; 95% CI, 1.04–5.80), colorectal (RR = 2.00; 95% CI, 1.24–3.22), breast (RR = 1.43; 95% CI, 1.13–1.81), female reproductive (RR = 1.66; 95% CI, 1.02–2.70), multiple myeloma (RR = 2.80; 95% CI, 1.34–5.83), and unspecified or other (RR = 1.99; 95% CI, 1.25–3.16). Psychological

**Table 3 | Secondary analyses**

| | UK Biobank | | | | Finnish Public Sector Study | | | |
|---|---|---|---|---|---|---|---|---|
| | All-cause mortality | | | | All-cause mortality | | | |
| | N deaths | N total | HR (95% CI) | p value | N deaths | N total | HR (95% CI) | p value |
| 1. Distress before cancer | | | | | | | | |
| Model 1 | 3460 | 13435 | 1.51 (1.31–1.73) | <0.001*** | 834 | 4382 | 1.04 (0.88–1.22) | 0.667 |
| Model 2 | 3460 | 13435 | 1.31 (1.13–1.51) | <0.001*** | 834 | 4382 | 1.02 (0.87-1.21) | 0.777 |
| 2. Participants without cancer | | | | | | | | |
| Model 1 | 15285 | 362087 | 2.36 (2.23–2.49) | <0.001*** | 2913 | 142562 | 1.17 (1.08–1.27) | <0.001*** |
| Model 3 | 15285 | 362087 | 1.66 (1.57–1.76) | <0.001*** | 2913 | 142562 | 1.12 (1.03–1.21) | 0.008** |

Cox proportional hazards models (two-tailed).
**$p < 0.01$, ***$p < 0.001$.
CI = 95% confidence interval.
Reference category = low distress.
Model 1 adjusted for age and sex.
Model 2 adjusted for age, sex, ethnicity, education, number of comorbidities, age at cancer diagnosis, time between cancer diagnosis and depressive symptoms assessment, and antidepressant medication.
Model 3 adjusted for age, sex, ethnicity, education, number of comorbidities, and antidepressant medication.

distress was associated with a lower risk of cancer mortality in people with gastrointestinal cancer (excluding colorectal cancer) (RR = 0.64; 95% CI; 0.42–0.96).

Meta-analyses of the results from the secondary analyses showed that psychological distress experienced 1–4 years prior to cancer diagnosis was associated with increased risk of all-cause mortality (RR = 1.17; 95% CI: 1.05–1.31). Psychological distress was also associated with increased risk of all-cause mortality in people without a diagnosis of cancer (RR = 1.46; 95% CI: 1.39–1.52).

**Sensitivity analyses**

Sensitivity analyses conducted in UK Biobank yielded similar results to the main analyses (1) when distress was measured continuously rather than categorically (Supplementary Tables 1 and 2), (2) after excluding people who died within one year of baseline assessments to account for reverse causality (Supplementary Tables 3 and 4), (3) when self-reported comorbidities rather than comorbidities ascertained through hospital records was adjusted for (Supplementary Tables 5 and 6), (4) when time from diagnosis was used as the underlying time scale (Supplementary Tables 7 and 8), and (5) when excluding individuals diagnosed with non-melanoma skin cancer (Supplementary Tables 9 and 10). In analyses conducted in a sample with complete data on the exposure, covariates, mediators, and outcome, psychological distress was associated with an increased risk of all-cause mortality (HR 1.55; 95% CI, 1.22–1.97). Furthermore, the association with cancer mortality was directionally consistent but imprecisely estimated (HR 1.34; 95% CI, 0.99–1.82) (Supplementary Tables 11 and 12).

Sensitivity analyses conducted in FPS, excluding individuals with advanced-stage cancer, yielded similar results to the main analyses (Supplementary Tables 13 and 14).

## Discussion

This study found that psychological distress is associated with an increased risk of both all-cause and cancer mortality in people with LWBC. Associations persisted after extensive adjustment for demographic and clinical covariates, including cancer stage, and for a range of biological and behavioural mediators. Systematic inflammation explained part of the association in both cohorts, whereas evidence for mediation through smoking, poor sleep, and physical inactivity was less consistent. We also found that the link between psychological distress and survival differed by cancer type, with particularly strong associations evident in individuals with head and neck, colorectal, non-

melanoma skin, breast, female reproductive, multiple myeloma, and unspecified or other cancers.

The observed 1.5-fold increased risk of all-cause and cancer mortality associated with psychological distress in individuals with LWBC is slightly higher than those reported in previous meta-analyses (range: 1.24–1.40 for all-cause mortality and 1.21-1.39 for cancer mortality)[12,22,23]. We examined whether this discrepancy could be attributable to our use of a binary categorisation of psychological distress; however, comparable associations were observed when using continuous scores, suggesting that the measurement approach alone does not explain the stronger effect sizes. The robustness of our findings was further supported by sensitivity analyses excluding people who died within one year of baseline assessments to account for reverse causality, excluding those with non-melanoma skin cancer, and restricting analyses to participants with no missing data on the exposure, covariates, mediators, and outcomes. We also found that, even after adjusting for cancer stage, a proxy for cancer severity, the associations of psychological distress with both all-cause and cancer mortality persisted.

Associations between psychological distress and mortality in people LWBC were partially mediated by inflammation in both cohorts. This finding aligns with prior evidence showing that inflammation is linked to both depression[24,25] and cancer progression[15–17]. Notably, a recent cohort study of 394 patients with head and neck cancer found that systemic inflammation fully mediated the association between depression and survival[18]. The greater degree of mediation observed in that study may reflect its focus on a single cancer type, whereas our study combined multiple cancer types. Indeed, the associations between psychological distress and mortality, and the pathways through which psychological distress influences survival, may differ by cancer type. In our study, there was some evidence for a suggestive mediating role of physical inactivity, smoking, and sleep; however, these estimates were less consistent across cohorts. Given that physical activity, smoking, and sleep are all linked to inflammation[26–28], it is possible that a complex pathway exists between distress, health behaviours, inflammation, and survival in people with LWBC. However, due to inconsistencies, further research is needed to clarify these relationships. Importantly, inflammation may also act as a marker of cancer severity. Higher levels of CRP indicate greater immune system disturbances, suggesting that inflammation may function as a confounder rather than a mediator. Diet, alcohol consumption, and BMI appeared to contribute little to the distress-

mortality associations in our study. These findings are in line with previous research, which has shown that psychological distress is not associated with diet or alcohol consumption in people LWBC[19].

The mediators found in this study only contributed a moderate amount to the association between distress and mortality, suggesting that other factors, such as treatment adherence, may play a role. Research shows that depression predicts non-adherence to medication in people with cancer[29,30] due to factors such as lack of motivation, decreased energy, or feelings of hopelessness[31]. Furthermore, non-adherence to medication has been associated with a greater risk of mortality in people with cancer[32–34]. Additionally, distress might affect other aspects of treatment, such as attendance at follow-up appointments or participation in clinical trials, which might also affect survival outcomes. Another potential mediator is tumour response to treatment. Psychological distress can lead to inflammation, which may affect how tumours respond to treatment, ultimately influencing cancer progression and survival[35]. Future research is warranted to further explore the mechanisms through which psychological distress affects survival among people with LWBC.

Stratified analyses revealed that the associations between distress and survival varied by cancer type. However, these patterns were not entirely consistent across the two cohorts. For example, in the UK Biobank, distress was associated with increased all-cause mortality among individuals with breast cancer, non-melanoma skin cancer, and multiple myeloma, whereas no such associations were observed in the FPS cohort. These differences may be partially attributable to differences in participant characteristics and cancer type distributions. Although the absolute number of breast cancer cases was similar in both cohorts, breast cancer accounted for a larger proportion of cancer cases in FPS (36.8% vs 19.6%). This likely reflects the higher proportion of women in FPS (82.8% vs 51.2%), as the sample is drawn from municipal employees in Finland, particularly in healthcare, education, and social services. These sectors are predominantly staffed by women. In contrast, non-melanoma skin cancer was more prevalent in the UK Biobank cohort (29.5% vs 12.5%). Furthermore, FPS participants had generally higher levels of education and were more likely to experience psychological distress. Additionally, a greater proportion of UK Biobank participants died before the end of follow-up compared to those in FPS (18.1% vs 8.8%), which may have influenced the observed associations.

Other meta-analyses have also found that the association between distress and mortality in people with LWBC may differ by cancer type[12,22,36]. Consistent with our results, associations between distress and poorer survival have been demonstrated in head and neck[37], breast[22], and colorectal cancer[36]. Furthermore, research suggests that psychological factors may have a smaller impact on cancer types that typically follow a predictable pattern of progression[11]. Individuals with head and neck cancers may be particularly vulnerable to psychological distress due to the disease itself, causing difficulty with swallowing, breathing, and speaking, as well as treatment side effects such as dry mouth and changes in appearance[38]. These factors may amplify the influence of distress on survival. In contrast, cancers with more stable trajectories, such as testicular cancer, may be less influenced by psychological factors, due to the relative predictability of treatment and disease progression. Together, these findings suggest that the mechanisms linking psychological distress to cancer outcomes may vary by cancer type. Future research should focus on further exploring the association between distress and mortality, as well as the underlying mechanisms, in specific cancer types.

This study has several strengths, including its large-scale multi-cohort design and validation in an independent population. Moreover, the sample includes participants with an array of different cancer types, enabling the analysis of the association between distress and mortality to be stratified by cancer type. We also applied two mediation approaches, the percentage of attenuation method and the inverse odds ratio-weighted method, allowing for methodological triangulation that enhances the robustness of our findings.

However, our results should also be interpreted in light of various limitations. First, we lacked data on cancer treatment. Second, we did not have repeated measures on psychological distress, limiting our ability to assess how changes in distress over time affect survival. While we excluded participants who died within one year of baseline to address reverse causality, the possibility that undiagnosed cancer progression could cause both distress and mortality remains. Longitudinal assessments of distress would have strengthened the analysis by allowing investigation of how changes in distress over time influence cancer progression and survival. Future work is warranted to investigate the association between trajectories of distress and survival to identify optimal timepoints for intervention. Third, we only had baseline data for the mediators, meaning that we could not explore how changes in the mediators over time might influence the distress-survival relationship. Future research using longitudinal data would be necessary to better understand the temporal dynamics of these relationships. Fourth, our focus on a single inflammatory marker, C-reactive protein, may limit the interpretation of the findings, as inflammation is a complex process. Future work could incorporate additional systemic biomarkers of inflammation, such as tumour necrosis factor alpha or interleukin-6, to provide a more comprehensive understanding of inflammatory pathways. Fifth, psychological distress was assessed using non-cancer-specific scales, which may not fully capture the unique distress experienced by people with LWBC. Sixth, the methods for assessing mediators varied between cohorts and relied largely on self-report. This reliance on self-report may introduce bias due to recall and social desirability biases, affecting the accuracy of the results. However, previous research shows that the self-report diet questions in the UK Biobank reliably rank participants according to intakes of the main food groups[39]. Similarly, the International Physical Activity Questionnaire (IPAQ), used to assess physical activity in the UK Biobank, has been shown to be a valid tool[40]. Nonetheless, future research could benefit from incorporating objective measures of physical activity, such as accelerometers, and biomarkers for diet to reduce potential bias and improve the validity of the findings.

In conclusion, this multicohort study showed that psychological distress is associated with an increased risk of all-cause and cancer mortality in people with LWBC. These associations persisted even after adjusting for cancer stage and were partly explained by higher systemic inflammation, while evidence for mediation by health behaviours was inconsistent. We also found that the association between distress and mortality among people with LWBC differs by cancer type. These findings underscore the importance of routinely screening for and treating psychological distress in people with LWBC. Future research with a wider range of potential mediators is needed to further explore the mechanisms through which distress may affect survival among people with LWBC.

## Methods

### Sex and gender reporting

In both the UK Biobank and the FPS, sex was recorded at baseline by self-report. No data on gender identity was collected. The UK Biobank sample included 13,349 participants, of whom 6836 were female (51.2%) and 6513 were male (48.8%). The FPS sample included 5739 participants, of whom 4752 were female (82.8%) and 987 were male (17.2%). Sex was included as a covariate in all models. Analyses were not stratified by sex as the aim of the study was to examine overall associations between psychological distress and mortality rather than sex-specific associations.

### Study design and population

This prospective, observational, multi-cohort study used data from the UK Biobank for the primary analysis[41] and the FPS for validation[42]. UK

Biobank is a large prospective cohort study that identified participants via the UK National Health Service (NHS) records. Of the 9.1 million adults eligible for inclusion in UK Biobank, 502,665 adults aged 38–73 years participated in a baseline clinical examination between 2006 and 2010. FPS is a prospective cohort study of public sector personnel in 11 towns and five well-being services counties in Finland. FPS includes individuals who responded to surveys during the periods 2000–2002, 2004–2005, 2008–2009, 2012–2013, and 2016–2017. For the present study, we included participants diagnosed with cancer, with the first diagnosis occurring four years or less before baseline assessments, consistent with previous studies[43].

## Cancer at baseline
Data on cancer type and date of diagnosis were obtained from national cancer registries. Cancer type was coded according to the 9th and 10th revisions of the International Classification of Diseases (ICD-9 and ICD-10) (Supplementary file). In FPS, data on cancer stage were also obtained from national registries.

## Psychological distress
In the UK Biobank, psychological distress was measured at baseline using the 4-item version of the Patient Health Questionnaire (PHQ-4)[44]. The PHQ-4 asks participants about the frequency of low mood, anhedonia, feelings of anxiety, and inability to control worrying over the past two weeks. Responses are rated on a four-point Likert scale from 0 "not at all" to 3 "nearly every day". The total score ranges from 0-12, with higher scores indicating greater distress. The established cut-off score of 6 or more was used to denote psychological distress[45,46]. The PHQ-4 has been shown to be a valid measure of psychological distress in people diagnosed with cancer[47,48].

In FPS, distress was measured at baseline using the 12-item General Health Questionnaire (GHQ-12)[49], which assesses the severity of mental distress over the past few weeks. Items are scored on a 4-point Likert scale (from 0-3), with higher scores indicating worse mental health. The total score ranges from 0-36. Based on the optimal cut-off point of 3 established in a Finnish validation study[50], participants were categorised as non-distressed (GHQ scores 0-3) and distressed (GHQ scores 4–12). The GHQ-12 has been shown to be a valid measure of psychological distress in people LWBC[51,52].

## Mediators
Mediators assessed at baseline, including alcohol consumption, physical activity, smoking, BMI, sleep duration, and sleep quality, were assessed in both cohorts, while data on inflammation (high-sensitivity C-reactive protein concentration in serum) and dietary intake (fruit, vegetable, red meat, and processed meat) were measured only in the UK Biobank. Table 4 summarises the mediators assessed in the UK Biobank and FPS, along with their methods of ascertainment. In addition, Supplementary File 1 provides full details of how all mediators were assessed.

## Covariates
Covariates included self-reported age, sex, ethnicity (dichotomised into White/non-White), education (low/intermediate/high), number of comorbidities, cancer stage (FPS only), age at cancer diagnosis, time between diagnosis and baseline assessments, and antidepressant use (yes/no). Comorbidities were assessed using hospitalisation data coded according to the ICD-10, including diabetes, Alzheimer's disease, asthma, dementia, Parkinson's disease, hypertension, angina, heart attack, stroke, heart failure, heart murmur, abnormal heart rhythm, and chronic kidney disease, categorised as 0 versus 1+. Date of diagnosis was used to determine age at cancer diagnosis and time between diagnosis and baseline assessments. In the UK Biobank, antidepressant use was assessed through self-reported medications (field 20003). A list of antidepressants developed in prior UK Biobank

studies was used to derive a variable indicating if participants had taken at least one antidepressant medication[53]. In FPS, antidepressant use was determined from the Medication Register/Social Insurance Institution of Finland (Anatomical Therapeutic Chemical code N06A), indicating if participants had purchased an antidepressant medication or not.

## Cancer and all-cause mortality
Data on the cause and date of death were obtained from national registries through linkage using participants' unique identification numbers from baseline to 31st March 2021 in the UK Biobank and 31st December 2018 in FPS. Causes of mortality were classified according to the ICD-10 and categorised into cancer-specific (C00-C99, D00-D49) and all-cause mortality.

## Statistics & reproducibility
We used Stata (Version 17) for UK Biobank analyses, SAS (Version 9.4) for FPS analyses, and R (Version 4.4.2) for meta-analyses.

**Missing data.** In the UK Biobank, missing data on predictors and covariates were assumed to be missing at random and imputed using multiple imputation by chained equations (MICE)[54]. Twenty datasets were imputed, and estimates were pooled using Rubin's rules (Rubin, 2004). The imputation model included all variables used in the main analyses (exposure, covariates, mediators, and outcomes), as well as auxiliary variables likely related to missingness (income). The proportion of missing data for key variables ranged from 0% (e.g., age, sex, number of comorbidities) to approximately 24% (e.g., physical activity).

**Main analyses.** In both datasets, we performed Cox proportional hazards models to examine the prospective association between distress and all-cause mortality. Competing risk regression based on Fine and Gray's proportional subhazard model[55] was used to test associations between psychological distress and cancer mortality[56], as this method accounts for competing causes of death that may preclude death from cancer. Survival time was measured in years from the time of baseline assessments to the date of death, or to the end of follow-up. Effect estimates were adjusted for (1) age and sex (minimally adjusted model), (2) age, sex, sociodemographic factors, age at cancer diagnosis, time between cancer diagnosis and depressive symptoms assessment, and antidepressant use (multivariable adjusted model). In FPS, a third model was fitted with additional adjustment for cancer stage.

To assess the potential role of the mediators in the association between distress and mortality, these were added separately as an additional covariate to the multivariable adjusted model (Model 2). The attenuation of excess risk after additional adjustment for the mediators was calculated using the following formula: Percentage of attenuation = $[(HR_1-HR_2)/(HR_1-1)]*100$, where $HR_1$ is the HR adjusted for confounders, and $HR_2$ represents the HR adjusted for confounders and the respective mediator.

To account for mediation involving multiple interrelated mediators simultaneously, we also applied an inverse odds ratio-weighted method[57]. This approach decomposes the total association between psychological distress and mortality (total effect) into a natural direct effect (NDE), not operating through the mediators, and a natural indirect effect (NIE), operating through the mediators jointly. In FPS, this analysis was conducted using both the multivariable adjusted model (Model 2) and a further model adjusting for cancer stage (Model 3).

**Secondary analyses.** To examine potential reverse associations (the association of psychological distress with subsequent cancer) in both datasets, we assessed whether (1) psychological distress experienced

**Table 4 | Mediators assessed in the UK Biobank and Finnish Public Sector Study**

| Mediator | Method of assessment in the UK Biobank | Method of Assessment in Finnish Public Sector Study |
|---|---|---|
| Inflammation | Serum C-reactive protein levels | Not available |
| Fruit intake | Touchscreen questionnaire | Not available |
| Vegetable intake | Touchscreen questionnaire | Not available |
| Red meat intake | Touchscreen questionnaire | Not available |
| Processed meat intake | Touchscreen questionnaire | Not available |
| Alcohol consumption | Touchscreen questionnaire | Self-reported |
| Physical activity | Modified version of the International Physical Activity Questionnaire (IPAQ) [34] | Self-reported |
| Smoking status | Touchscreen questionnaire | Self-reported |
| Body mass index | Nurse assessed height and weight | Self-reported height and weight |
| Sleep duration | Touchscreen questionnaire | Self-reported |
| Sleep quality | Touchscreen questionnaire | Jenkins Sleep Problem Scale [35] |

1–4 years **before** a cancer diagnosis (as opposed to 4 years **after** a cancer diagnosis in the main analyses) was associated with all-cause mortality in people LWBC, and (2) whether psychological distress was associated with all-cause mortality among participants without cancer.

**Meta-analysis.** Where possible, we pooled study-specific effect estimates from the UK Biobank and FPS using fixed-effects meta-analysis. To further examine whether the associations between psychological distress and mortality vary across cancer type, we conducted analyses stratified by cancer type.

**Sensitivity analyses.** Additional sensitivity analyses were carried out in our sample of UK Biobank participants (primary analysis). First, models were re-estimated using a continuous measure of psychological distress instead of a binary classification. Second, participants who died within one year of baseline assessment were excluded to reduce the potential of reverse causality. Third, the analyses were repeated using self-reported comorbidities instead of hospital records. These included diabetes, Alzheimer's, asthma, dementia, Parkinson's, hypertension, angina, heart attack, stroke, heart failure, heart murmur, abnormal heart rhythm, and chronic kidney disease. The total number of comorbidities was categorised as 0 versus >1. Fourth, analyses were repeated in a complete-case sample with no missing data on the exposure, covariates, mediators, and outcomes. Fifth, models were rerun using time since diagnosis (rather than time since baseline) as the underlying time scale. Sixth, analyses were repeated excluding individuals with non-melanoma skin cancer.

In our validation cohort of FPS participants, we conducted an additional sensitivity analysis excluding individuals with advanced-stage cancer.

## Ethics
This research complies with all relevant ethical regulations. UK Biobank was approved by the National Health Service National Research Ethics Service. The present study was conducted using the UK Biobank Resource (Application numbers 60565). FPS was approved by the Helsinki Uusimaa Hospital District Ethics Committee (HUS/1210/2016). All participants provided written informed consent prior to participation in data collection.

## Reporting summary
Further information on research design is available in the Nature Portfolio Reporting Summary linked to this article.

## Data availability
Researchers registered with the UK Biobank can apply for access to the database by completing an application. This must include a summary of the research plan, data fields required, any new data or variables that will be generated, and payment to cover the incremental costs of servicing an application (https://www.ukbiobank.ac.uk/enable-your-research/apply-for-access). In the Finnish Public Sector study, pseudonymised questionnaire data can be shared upon request to the investigators. Linked health records require separate permission from the Findata, the Health and Social Data Permit Authority in Finland. Source data are provided with this paper.

## Code availability
Code for UK Biobank analyses can be found at the following website https://codeocean.com (https://doi.org/10.24433/CO.3155775.v1).

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

## Acknowledgements

NM is funded by the ESRC-BBSRC Soc-B Centre for Doctoral Training (ES/P000347/1). JP was supported by the Research Council of Finland (350426). MK was supported by Wellcome Trust, UK (221854/Z/20/Z), National Institute on Aging (NIH), US (R01AG056477), Medical Research Council, UK (MR/R024227/1, MR/Y014154/1), and Research Council of Finland (350426). During the conduct of the study, PF was supported by the Wellcome Trust (221854/Z/20/Z). For the purpose of Open Access, the author has applied a CC BY public copyright licence to any Author Accepted Manuscript version arising from this submission. The funders of the study had no role in study design, data collection, data analysis, data interpretation, or writing of the report.

## Author contributions

N.M., together with P.F. and A.F., designed the study. N.M. wrote the first draft. All authors participated in data interpretation and critically reviewed the report. N.M., J.P., and P.F. analysed the data. M.K. and J.P. were responsible for the data from FPS. N.M. and P.F. were responsible for the UKB data.

## Competing interests

The authors declare no competing interests.
