## [Transparent Peer Review file · Nature Communications]

Mediators of the association between psychological distress and mortality in people diagnosed with cancer

Corresponding Author: Ms Natalie Miller

Version 0:

Reviewer comments:

Reviewer #1

(Remarks to the Author)

This article utilizes two databases for analysis and presents a relatively novel perspective on the impact of psychological distress on survival after cancer. However, there are still several issues that need to be addressed and revised:

1. There are tables where the sum of the percentages is not 100%, such as in the Alive column of the FPS in Table 2, and the sum of the percentages for gender, as well as in Supplementary table 1. Authors are advised to check in full.
2. Please summarize the hazard ratios (HRs) after adjustment for the mediating factor, along with the corresponding risk attenuation, in a tabular format. Please organize the logic of the results section and ensure that each textual explanation corresponds precisely to the appropriate figure or table.
3. Key factors such as cancer stage, and treatment modality may exert a crucial influence on the observed association between psychological distress and survival outcomes.
4. Several aspects of the manuscript are misrepresented or inaccurately conveyed. The significance symbol for the multiple myeloma label in UKB in Table 3 is incorrect. The covariates listed in the supplementary table include marital status, which is inconsistent with the adjustments described in the main text. It is recommended that authors clarify the meaning of NA in the table.
5. It is recommended that the authors use appropriate mediation analysis tools to explore whether mediation effects exist and to assess the size of the mediation proportion. The author explores the mediating effects of variables such as inflammation, smoking, and alcohol consumption. However, when discussing the mediating effect of a particular factor, can the potential influence of the remaining factors on the results be considered and controlled for?

(Remarks on code availability)

Reviewer #2

(Remarks to the Author)

The authors of this paper aimed to examine the association between psychological distress and both cancer mortality and all-cause mortality in people LWBC, and to determine the extent to which inflammation and health behaviours may mediate these associations. Although they used two cohorts to assess this, I have several major concerns about this manuscript:

1. Both cohorts did not have information on tumor stage, which is most important for cancer mortality. Therefore, it is unknown whether the distress is just an indicator of cancer stage.
2. In the mediation analysis, the authors assessed diet, physical activity, alcohol consumption, smoking, sleep duration and sleep quality. All these might be causes of distress, and in a cross-sectional assessment at the baseline, it is not sure about the causality, which is quite important for mediation analysis.
3. The underlying time scale for the analysis is not clear. It should be time since diagnosis for cancer survival.
4. Taking a look at the details of different cancer types for the two cohorts, you will find that the results are not consistent between them in specific cancers.
5. Why Finnish Public Sector Study has so many females?

(Remarks on code availability)

Reviewer #3

(Remarks to the Author)

See the attachment.

(Remarks on code availability)

Version 1:

Reviewer comments:

Reviewer #1

(Remarks to the Author)

I have no more questions.

(Remarks on code availability)

Reviewer #2

(Remarks to the Author)

The authors have answered all my concerns. One additional question is that mediation analysis was performed for all cause mortality. Why the authors did not perform it for cancer specific mortality?

(Remarks on code availability)

Reviewer #3

(Remarks to the Author)

The author has addressed all the issues raised in my review comments, and I have no further comments.

(Remarks on code availability)

Reviewer #1:

This article utilizes two databases for analysis and presents a relatively novel perspective on the impact of psychological distress on survival after cancer. However, there are still several issues that need to be addressed and revised:

1. There are tables where the sum of the percentages is not 100%, such as in the Alive column of the FPS in Table 2, and the sum of the percentages for gender, as well as in Supplementary table 1. Authors are advised to check in full.

- Thank you for noticing this, we have checked this in all Tables and Supplementary Tables and have made the necessary edits to ensure all percentages add up to 100%. See below for examples of rows we have edited.

Table 2. Characteristics of individuals diagnosed with cancer within 4 years prior to baseline assessments in the UK Biobank and Replication Cohort (Finnish Public Sector Study)

	UK Biobank ^a			Finnish Public Sector Study		
	Total (N = 13349)	Died (N = 2421)	Alive (N = 10928)	Total (N = 5739)	Died (N = 505)	Alive (N = 5234)
	Mean (SD)/n (%)	Mean (SD)/n (%)	Mean (SD)/n (%)	Mean (SD)/n (%)	Mean (SD)/n (%)	Mean (SD)/n (%)
Sex						
Male	6513 (48.8)	1446 (59.7)	5861 (53.6)	987 (17.2)	100 (19.8)	887 (17.0)
Female	6836 (51.2)	975 (40.3)	5067 (46.4)	4752 (82.8)	405 (80.2)	4347 (83.0)
Cancer type						
Head and neck	215 (1.6)	58 (2.4)	157 (1.4)	70 (1.2)	7 (1.4)	63 (1.2)
Gastrointestinal (excluding colorectal)	253 (1.9)	117 (4.8)	136 (1.2)	71 (1.2)	24 (4.8)	47 (0.9)
Colorectal	983 (7.4)	321 (10.3)	733 (6.7)	252 (4.4)	32 (6.3)	220 (4.2)
Respiratory (including lung)	226 (1.7)	109 (4.5)	117 (1.1)	48 (0.8)	18 (3.6)	30 (0.6)
Bone, mesothelial and soft tissue	102 (0.8)	37 (1.5)	65 (0.6)	43 (0.8)	9 (1.8)	34 (0.7)
Melanoma	582 (4.4)	79 (3.3)	503 (4.6)	303 (5.3)	17 (3.4)	286 (5.5)
Non-melanoma skin	3950 (29.5)	392 (16.2)	3558 (32.6)	715 (12.5)	15 (3.0)	700 (13.4)
Breast	2624 (19.6)	405 (16.7)	2219 (20.3)	2112 (36.8)	183 (36.2)	1929 (36.9)
Female reproductive	641 (4.8)	116 (4.8)	525 (4.8)	397 (6.9)	49 (9.7)	348 (6.7)
Male reproductive	1925 (14.4)	372 (15.4)	1553 (14.2)	357 (6.2)	28 (5.5)	329 (6.3)
Urinary tract	400 (3.0)	126 (5.2)	274 (2.5)	145 (2.5)	18 (3.6)	127 (2.4)

Endocrine	108 (0.8)	14 (0.6)	94 (0.9)	209 (3.6)	8 (1.6)	201 (3.8)
Lymphoma	448 (3.4)	108 (4.5)	340 (3.1)	153 (2.7)	15 (3.0)	138 (2.6)
Multiple myeloma	110 (0.8)	65 (2.7)	45 (4.1)	51 (0.9)	10 (2.0)	41 (0.8)
Leukaemia	185 (1.4)	59 (2.4)	126 (1.2)	68 (1.2)	2 (0.4)	66 (1.3)
Unspecified or other (including eye, brain, neural)	597 (4.5)	113 (4.7)	483 (4.4)	745 (13.0)	70 (13.9)	675 (12.9)
Red meat intake				-	-	-
Less than three times a week	10148 (76.0)	1793 (74.1)	8355 (76.5)			
Three times a week or more	3201 (24.0)	628 (25.9)	2573 (23.5)			
Sleep duration						
Normal (7-9 hours)	9843 (73.7)	1675 (69.2)	8168 (74.7)	4206 (73.3)	372 (73.7)	3834 (73.2)
Short (<7 hours) or long (>9 hours)	3506 (26.3)	746 (30.8)	2760 (25.3)	1533 (26.7)	133 (26.3)	1400 (26.8)

Supplementary table 1. UK Biobank participant characteristics in observed and imputed data ($N = 13,349$)

	Observed Mean (SD)/n (%)	Imputed Mean (SD)/n (%) ^a
Cancer type, $N = 13,349$		
Head and neck	215 (1.6)	215 (1.6)
Gastrointestinal (excluding colorectal)	253 (1.9)	253 (1.9)
Colorectal	983 (7.4)	983 (7.4)
Respiratory (including lung)	226 (1.7)	226 (1.7)
Bone, mesothelial and soft tissue	102 (0.8)	102 (0.8)
Melanoma	582 (4.4)	582 (4.4)
Non-melanoma skin	3950 (29.5)	3950 (29.5)
Breast	2624 (19.6)	2624 (19.6)
Female reproductive	641 (4.8)	641 (4.8)
Male reproductive	1925 (14.4)	1925 (14.4)
Urinary tract	400 (3.0)	400 (3.0)
Endocrine	108 (0.8)	108 (0.8)
Lymphoma	448 (3.4)	448 (3.4)
Multiple myeloma	110 (0.8)	110 (0.8)
Leukaemia	185 (1.4)	185 (1.4)
Unspecified or other (including eye, brain, neural)	597 (4.5)	597 (4.5)
Red meat intake, $N = 13,207$		
Less than three times a week	10041 (76.0)	10149 (76.0)
Three times a week or more	3166 (24.0)	3200 (24.0)
BMI, $N = 13,274$		
Underweight or healthy weight	4351 (32.8)	4374 (32.8)

Overweight or obese

8923 (67.2)

8975 (67.2)

2. Please summarize the hazard ratios (HRs) after adjustment for the mediating factor, along with the corresponding risk attenuation, in a tabular format. Please organize the logic of the results section and ensure that each textual explanation corresponds precisely to the appropriate figure or table.

- We have added Tables showing the hazard ratios after adjustment for the mediating factor, along with the corresponding risk attenuation. These are Tables 4 and 6 (for all-cause and cancer specific mortality, respectively). We have added references to these in the text.

“In UK Biobank, psychological distress was associated with a 1.43 times increased risk of all-cause mortality after adjustment for all covariates (95% CI, 1.23-1.67) (Table 3). Additional adjustment for inflammation reduced the association by 18.6%, smoking 16.3%, sleep duration 9.3%, physical activity 7%, sleep quality 2.3%, alcohol consumption 2.3%, and fruit intake 2.3% (Table 4).” Page 12.

“In FPS, the results were consistent with those observed in the UK Biobank. Psychological distress was associated with a 1.72 times increased risk of all-cause mortality after multivariable adjustment (95% CI: 1.44-2.07) (Table 3). Additional adjustment for physical activity reduced the association by 5.6%, sleep quality 2.8% and smoking 1.4% (Table 4).” Page 12

Table 4. Hazard ratios for association between psychological distress and all-cause mortality after adjustment for each mediator.

Mediator (Model 3)	UK Biobank		FPS	
	Adjusted HRs (95% CI)	% of attenuation	Adjusted HRs (95% CI)	% of attenuation
Inflammation	1.35 (1.15-1.57)	18.6	-	-
Fruit	1.42 (1.22-1.66)	2.3	-	-
Vegetables	1.43 (1.23-1.67)	0.0	-	-
Red meat	1.43 (1.23-1.67)	0.0	-	-
Processed meat	1.43 (1.23-1.67)	0.0	-	-
Physical activity	1.40 (1.20-1.63)	7.0	1.68 (1.40-2.02)	5.6
BMI	1.44 (1.23-1.67)	0.0	1.73 (1.44-2.08)	0.0
Alcohol	1.42 (1.22-1.66)	2.3	1.73 (1.44-2.08)	0.0
Smoking	1.36 (1.17-1.59)	16.3	1.71 (1.43-2.06)	1.4
Sleep quality	1.42 (1.21-1.65)	2.3	1.70 (1.40-2.06)	2.8
Sleep duration	1.39 (1.19-1.62)	9.3	1.75 (1.45-2.11)	0.0

All models adjusted for age, sex, ethnicity, education, number of comorbidities, age at cancer diagnosis, time between cancer diagnosis and depressive symptoms assessment, and antidepressant medication (yes/no).

“In UK Biobank, psychological distress was associated with a 1.45-fold increased risk of cancer mortality after adjustment for all covariates (95% CI, 1.21-1.74) (Table 5). Additional adjustment for inflammation reduced the association by 17.8%, smoking 13.3%, sleep duration 11.1%, physical activity 6.7%, sleep quality 2.2%, alcohol 2.2% and fruit consumption 2.2% (Table 6).” Page 12

“In FPS, distress was also associated with a 1.72-fold increased risk of cancer mortality after multivariable adjustment (95% CI, 1.41-2.10) (Table 5). Additional adjustment for sleep quality reduced the association by 8.3% and physical activity by 5.6%. Alcohol, smoking, BMI and sleep duration did not attenuate the association (Table 6).” Page 13

Table 6. Hazard ratios for association between psychological distress and cancer mortality after adjustment for each mediator.

Mediator (Model 3)	UK Biobank		FPS	
	Adjusted SHRs (95% CI)	% of attenuation	Adjusted SHRs (95% CI)	% of attenuation
Inflammation	1.37 (1.14-1.64)	17.8	-	-
Fruit	1.44 (1.20-1.72)	2.2	-	-
Vegetables	1.45 (1.21-1.74)	0.0	-	-
Red meat	1.45 (1.21-1.74)	0.0	-	-
Processed meat	1.45 (1.21-1.74)	0.0	-	-
Physical activity	1.42 (1.18-1.70)	6.7	1.68 (1.38-2.05)	5.6
BMI	1.45 (1.21-1.74)	0.0	1.72 (1.41-2.11)	0.0
Alcohol	1.44 (1.20-1.73)	2.2	1.73 (1.41-2.11)	0.0
Smoking	1.39 (1.16-1.67)	13.3	1.72 (1.40-2.10)	0.0
Sleep quality	1.44 (1.20-1.73)	2.2	1.66 (1.35-2.04)	8.3
Sleep duration	1.40 (1.20-1.69)	11.1	1.75 (1.43-2.14)	0.0

All models adjusted for age, sex, ethnicity, education, number of comorbidities, age at cancer diagnosis, time between cancer diagnosis and depressive symptoms assessment, and antidepressant medication (yes/no).

3. Key factors such as cancer stage, and treatment modality may exert a crucial influence on the observed association between psychological distress and survival outcomes.

- The UK Biobank has not released data on cancer stage; however, we have conducted two additional analyses in the Finnish Public Sector (FPS) study (1) including cancer stage as an additional covariate in the association between psychological distress and all-cause/cancer mortality and (2) a sensitivity analysis excluding people with advanced stage cancer.
- These analyses found that cancer stage attenuates but does not remove the association between distress and all-cause mortality.
- We have added the methods for these analyses on Pages 9, 10 and 11, descriptive statistics for cancer stage in Table 2, the results on Pages 12, 13 and 15 and in Tables 4 and 6, as well as Supplementary Tables 14 and 15, and a brief discussion of the findings on Page 16. Unfortunately, neither cohort study has data on cancer treatments, and we acknowledge this as a limitation of our work in the ‘Strengths and Limitations’ section of the Discussion (Page 19).

“Effect estimates were adjusted for (1) age and sex (minimally adjusted model), (2) age, sex, sociodemographic factors, age at cancer diagnosis, time between cancer diagnosis and depressive symptoms assessment, and antidepressant use (multivariable adjusted model). In FPS, a third model was fitted with additional adjustment for cancer stage.” Pages 9/10

“In FPS, the results were consistent with those observed in the UK Biobank. Psychological distress was associated with a 1.72 times increased risk of all-cause mortality after multivariable adjustment (95% CI: 1.44-2.07) (Table 3). Additional adjustment for cancer stage attenuated the association by 22%, although the relationship remained statistically significant (Table 4).” Page 12

“In UK Biobank, psychological distress was associated with a 1.45-fold increased risk of cancer mortality after adjustment for all covariates (95% CI, 1.21-1.74) (Table 5). Additional adjustment for cancer stage attenuated the association by 25%, although the relationship remained statistically significant (Table 6).” Page 12/13

“In the FPS, additional adjustment for cancer stage to account for cancer severity led to substantial attenuation of associations between psychological distress and all-cause or cancer mortality, although the relationships remained.” Page 16

“First, we lacked data on cancer treatment. This meant we were unable to account for the potential impact of cancer treatments on associations between distress and survival. Additionally, we could not account for the potential impact of cancer treatments on the immune system when exploring inflammation as a potential mediator.” Page 19

Table 2. Characteristics of individuals diagnosed with cancer within 4 years prior to baseline assessments in the UK Biobank and Replication Cohort (Finnish Public Sector Study)

UK Biobank ^a			Finnish Public Sector Study		
Total (N = 13349)	Died (N = 2421)	Alive (N = 10928)	Total (N = 5739)	Died (N = 505)	Alive (N = 5234)

	Mean (SD)/n (%)	Mean (SD)/n (%)	Mean (SD)/n (%)	Mean (SD)/n (%)	Mean (SD)/n (%)	Mean (SD)/n (%)
Cancer stage	-	-	-			
Unknown				1512 (26.4)	64 (12.7)	1448 (27.7)
Localised				2779 (48.4)	149 (29.5)	2630 (50.3)
Non-localised, only regional lymph node metastases				821 (14.3)	113 (22.4)	708 (13.5)
Metasised farther than to regional lymph nodes or invades adjacent tissues				274 (4.8)	88 (17.4)	186 (3.6)
Non-localised, no information on extent				167 (2.9)	35 (6.9)	132 (2.5)
Locally advanced, tumour invades adjacent tissues				68 (1.2)	11 (2.2)	57 (1.1)
Non-localised, also distant lymph node metastases				118 (2.1)	45 (8.9)	73 (1.4)

Table 4. Hazard ratios for association between psychological distress and all-cause mortality after adjustment for each mediator.

	UK Biobank		FPS	
Model 2	Adjusted HRs (95% CI)	-	Adjusted HRs (95% CI)	-
	1.43 (1.23-1.67)	-	1.72 (1.44-2.07)	-
Model 2 + cancer stage	Adjusted HRs (95% CI)	-	Adjusted HRs (95% CI)	% of attenuation
	-	-	1.56 (1.29-1.88)	22.2
Model 2 + mediator	Adjusted HRs (95% CI)	% of attenuation	Adjusted HRs (95% CI)	% of attenuation
Inflammation	1.35 (1.15-1.57)	18.6	-	-
Fruit	1.42 (1.22-1.66)	2.3	-	-
Vegetables	1.43 (1.23-1.67)	0.0	-	-
Red meat	1.43 (1.23-1.67)	0.0	-	-
Processed meat	1.43 (1.23-1.67)	0.0	-	-
Physical activity	1.40 (1.20-1.63)	7.0	1.68 (1.40-2.02)	5.6
BMI	1.44 (1.23-1.67)	0.0	1.73 (1.44-2.08)	0.0
Alcohol	1.42 (1.22-1.66)	2.3	1.73 (1.44-2.08)	0.0
Smoking	1.36 (1.17-1.59)	16.3	1.71 (1.43-2.06)	1.4
Sleep quality	1.42 (1.21-1.65)	2.3	1.70 (1.40-2.06)	2.8
Sleep duration	1.39 (1.19-1.62)	9.3	1.75 (1.45-2.11)	0.0

All models adjusted for age, sex, ethnicity, education, number of comorbidities, age at cancer diagnosis, time between cancer diagnosis and depressive symptoms assessment, and antidepressant medication (yes/no).

Table 6. Hazard ratios for association between psychological distress and cancer mortality after adjustment for each mediator.

	UK Biobank		FPS	
Model 2	Adjusted HRs (95% CI)	-	Adjusted HRs (95% CI)	-
	1.45 (1.21-1.74)	-	1.72 (1.41-2.10)	-
Model 2 + cancer stage	Adjusted HRs (95% CI)	-	Adjusted HRs (95% CI)	% of attenuation
	-	-	1.54 (1.26-1.90)	25.0
Model 2 + mediator	Adjusted SHRs (95% CI)	% of attenuation	Adjusted SHRs (95% CI)	% of attenuation
Inflammation	1.37 (1.14-1.64)	17.8	-	-
Fruit	1.44 (1.20-1.72)	2.2	-	-
Vegetables	1.45 (1.21-1.74)	0.0	-	-
Red meat	1.45 (1.21-1.74)	0.0	-	-
Processed meat	1.45 (1.21-1.74)	0.0	-	-
Physical activity	1.42 (1.18-1.70)	6.7	1.68 (1.38-2.05)	5.6
BMI	1.45 (1.21-1.74)	0.0	1.72 (1.41-2.11)	0.0
Alcohol	1.44 (1.20-1.73)	2.2	1.73 (1.41-2.11)	0.0
Smoking	1.39 (1.16-1.67)	13.3	1.72 (1.40-2.10)	0.0
Sleep quality	1.44 (1.20-1.73)	2.2	1.66 (1.35-2.04)	8.3
Sleep duration	1.40 (1.20-1.69)	11.1	1.75 (1.43-2.14)	0.0

All models adjusted for age, sex, ethnicity, education, number of comorbidities, age at cancer diagnosis, time between cancer diagnosis and depressive symptoms assessment, and antidepressant medication (yes/no).

Supplementary table 14. Association between psychological distress and all-cause mortality risk, additionally adjusting for cancer stage (FPS).

All-cause mortality		
	HR (95% CI)	p value
N deaths/N total = 505/5739		
Model 1	1.80 (1.50-2.16)	<0.001***
Model 2	1.72 (1.44-2.07)	<0.001***
N deaths/N total = 326/5112		
Model 2 (excluded stages 3-6)	1.56 (1.24, 1.97)	0.0002**
N deaths/N total = 213/4291		
Model 2 (excluded stages 2-6)	1.71 (1.28, 2.28)	0.0002**

Notes. **p*<0.05; ***p*<0.01, ****p*<0.001.

CI = confidence intervals, HR = hazard ratio.

Model 1 adjusted for age and sex.

Model 2 adjusted for age, sex, ethnicity, education, number of comorbidities, age at cancer diagnosis, time between cancer diagnosis and depressive symptoms assessment, and antidepressant medication (yes/no).

Supplementary table 15. Association between psychological distress and cancer mortality risk, additionally adjusting for cancer stage (FPS).

	Cancer mortality	
	SHR (95% CI)	p value
N deaths/N total = 440/5739		
Model 1	1.70 (1.47, 2.18)	<0.001***
Model 2	1.72 (1.41, 2.10)	<0.001***
N deaths/N total = 269/5112		
Model 2 (excluded stages 3-6)	1.57 (1.21, 2.03)	0.0007**
N deaths/N total = 169/4291		
Model 2 (excluded stages 2-6)	1.76 (1.28, 2.43)	0.0006**

Notes. * $p < 0.05$; ** $p < 0.01$, *** $p < 0.001$.

CI = confidence intervals, HR = hazard ratio.

Model 1 adjusted for age and sex.

Model 2 adjusted for age, sex, ethnicity, education, number of comorbidities, age at cancer diagnosis, time between cancer diagnosis and depressive symptoms assessment, and antidepressant medication (yes/no).

4. Several aspects of the manuscript are misrepresented or inaccurately conveyed. The significance symbol for the multiple myeloma label in UKB in Table 3 is incorrect. The covariates listed in the supplementary table include marital status, which is inconsistent with the adjustments described in the main text. It is recommended that authors clarify the meaning of NA in the table.

- Thank you for noticing these errors. We have adjusted the significance symbol for the multiple myeloma label in Table 3. We have also clarified the meaning of N/A – which was that the data were insufficient for the analysis to run.
- Additionally, we have adjusted the covariates listed in all supplementary tables to match what was actually adjusted for (not marital status).

Table 3. Association between psychological distress and all-cause mortality risk among people LWBC.

	UK Biobank				Finnish Public Sector Study			
	All-cause mortality				All-cause mortality			
	N deaths	N total	HR (95% CI)	p value	N deaths	N total	HR (95% CI)	p value
Model 1	2421	13349	1.69 (1.45-1.96)	<0.001***	505	5739	1.80 (1.50-2.16)	<0.001***
Model 2	2421	13349	1.43 (1.23-1.67)	<0.001***	505	5739	1.72 (1.44-2.07)	<0.001***
Cancer type (Model 2)								
Head and neck	58	215	2.16 (0.93-5.02)	0.073	7	70	1.88 (0.21-16.67)	0.570
Gastrointestinal (excluding colorectal)	117	253	0.56 (0.27-1.19)	0.122	24	71	2.08 (0.78-5.53)	0.143
Colorectal	321	983	1.61 (0.95-2.75)	0.078	32	252	2.55 (1.21-5.38)	0.014*
Respiratory (including lung)	109	226	1.11 (0.54-2.28)	0.780	18	48	1.68 (0.62-4.56)	0.307
Bone, mesothelial and soft tissue	37	102	1.53 (0.39-5.99)	0.544	9	43	1.46 (0.16-13.80)	0.741
Melanoma	79	582	1.04 (0.33-3.34)	0.945	17	303	1.07 (0.34-3.40)	0.903

Non-melanoma skin	392	3950	2.01 (1.34-3.02)	0.001**	15	715	0.64 (0.13-3.21)	0.590
Breast	405	2624	1.55 (1.12-2.14)	0.009**	183	2112	1.26 (0.92-1.71)	0.146
Female reproductive	116	641	1.28 (0.62-2.62)	0.507	49	397	1.87 (1.04-3.38)	0.036*
Male reproductive	372	1925	1.19 (0.76-1.87)	0.447	28	357	2.27 (0.98-5.28)	0.057
Urinary tract	126	400	1.70 (0.84-3.43)	0.139	18	145	1.90 (0.62-5.85)	0.263
Endocrine	14	108	3.47 (0.54-22.1)	0.188	8	209	N/A (insufficient data)	
Lymphoma	108	448	1.72 (0.86-3.45)	0.127	15	153	1.69 (0.56-5.12)	0.351
Multiple myeloma	65	110	3.42 (1.22-9.56)	0.019*	10	51	0.43 (0.08-2.31)	0.323
Leukaemia	59	185	0.99 (0.36-2.71)	0.986	2	68	N/A (insufficient data)	
Unspecified or other (including eye, brain, neural)	113	597	0.66 (0.27-1.60)	0.361	70	745	2.43 (1.49-3.96)	<0.001***

Notes. *p<0.05; **p<0.01, ***p<0.001.

CI = confidence intervals, HR = hazard ratio.

Reference category = low distress.

Model 1 adjusted for age and sex.

Model 2 adjusted for age, sex, ethnicity, education, number of comorbidities, age at cancer diagnosis, time between cancer diagnosis and depressive symptoms assessment, and antidepressant medication (yes/no).

Example of a Supplementary table (this change has been made to all supplementary tables to reflect the correct number of covariates adjusted for):

Supplementary table 2. Association between psychological distress (continuous scores) and all-cause mortality risk over 15 years of follow-up among people LWBC (UK Biobank).

All-cause mortality

	HR (95% CI)	p value
N deaths /N total = 2421/13349		
Model 1	1.11 (1.09-1.13)	<0.001***
Model 2	1.09 (1.07-1.11)	<0.001***

Notes. * $p < 0.05$; ** $p < 0.01$, *** $p < 0.001$.

CI = confidence intervals, HR = hazard ratio.

Model 1 adjusted for age and sex.

Model 2 adjusted for age, sex, ethnicity, education, number of comorbidities, age at cancer diagnosis, time between cancer diagnosis and depressive symptoms assessment, and antidepressant medication (yes/no).

5. It is recommended that the authors use appropriate mediation analysis tools to explore whether mediation effects exist and to assess the size of the mediation proportion. The author explores the mediating effects of variables such as inflammation, smoking, and alcohol consumption. However, when discussing the mediating effect of a particular factor, can the potential influence of the remaining factors on the results be considered and controlled for?

- We thank the reviewer for their insightful suggestion to use appropriate mediation analysis tools to assess mediation while accounting for multiple mediators simultaneously.
- To address this, we have conducted an additional mediation analysis using the inverse odds ratio weighted method. This approach allows simultaneous consideration of multiple mediators to decompose the total effect into direct and indirect methods.
- We have included the methods for these additional analyses (Page 10), the results (Page 14), and an interpretation (Page 18) in the revised manuscript, which strengthens the robustness of our findings by methodological triangulation (Page 20)

“To account for mediation involving multiple interrelated mediators simultaneously, we also applied an inverse odds ratio-weighted method specifically for the outcome of all-cause mortality³⁸. This approach decomposes the total association between psychological distress and mortality (total effect) into a natural direct effect (NDE), not operating through the mediators, and a natural indirect effect (NIE), operating through the mediators jointly. In FPS, this analysis was conducted using both the multivariable adjusted model (Model 2) and a further model adjusting for cancer stage (Model 3).” Page 10

“To account for mediation by multiple biological and behavioural factors simultaneously, we applied the inverse odds ratio-weighted method. In UK Biobank, this approach indicated that approximately 29% (95% CI: -46.9, 104.8) of the association between psychological distress and all-cause mortality was mediated by the combined set of mediators. In FPS, the combined mediators accounted for 25.7% (95% CI: 1.7-49.7) of the association in the multivariable adjusted model (Model 2). After further adjustment for cancer stage (Model 3), the mediation estimate was similar (25.6%) but was no longer statistically significant (95% CI: -6.4, 57.5).” Page 14

“The mediators found in this study only contributed a moderate amount to the association between distress and mortality, suggesting that other factors, such as treatment adherence, may play a role. Research shows that depression predicts non-adherence to medication in people with cancer^{46,47} due to factors such as lack of motivation, decreased energy, or feelings of hopelessness.⁴⁸ Furthermore, non-adherence to medication has been associated with greater risk of mortality in people with cancer.⁴⁹⁻⁵¹ Additionally, distress might affect other aspects of treatment, such as attendance at follow-up appointments or participation in clinical trials, which might also affect survival outcomes. Another potential mediator is tumour response to treatment. Psychological distress can lead to inflammation, which may affect how tumours respond to treatment, ultimately influencing cancer progression and survival.⁵² Future research is needed to further explore the mechanisms through which psychological distress affects survival among people LWBC.” Page 18

“We also applied two mediation approaches, the percentage of attenuation method and the inverse odds ratio-weighted method, allowing for methodological triangulation that enhances the robustness of our findings.” Page 20

Reviewer #2:

The authors of this paper aimed to examine the association between psychological distress and both cancer mortality and all-cause mortality in people LWBC, and to determine the extent to which inflammation and health behaviours may mediate these associations. Although they used two cohorts to assess this, I have several major concerns about this manuscript:

1. Both cohorts did not have information on tumor stage, which is most important for cancer mortality. Therefore, it is unknown whether the distress is just an indicator of cancer stage.

- The UK Biobank has not released data on cancer stage; however, we have conducted two additional analyses in the Finnish Public Sector (FPS) study (1) including cancer stage as an additional covariate in the association between psychological distress and all-cause/cancer mortality and (2) excluding people with advanced stage cancer.
- These analyses found that cancer stage attenuates but does not remove the association between distress and all-cause mortality.
- We have added the Methods for these analyses on Pages 9/10 and Page 11 (sensitivity analysis), descriptive statistics for cancer stage in Table 2, the results on Pages 12/13 and 15 and in Tables 4 and 6, as well as Supplementary Tables 14 and 15, and a brief Discussion of the findings on Page 16. Unfortunately, neither cohort study has data on cancer treatments, and we acknowledge this as a limitation of our work in the ‘Strengths and Limitations’ section of the Discussion (Page 19).

“Effect estimates were adjusted for (1) age and sex (minimally adjusted model), (2) age, sex, sociodemographic factors, age at cancer diagnosis, time between cancer diagnosis and depressive symptoms assessment, and antidepressant use (multivariable adjusted model). In FPS, a third model was fitted with additional adjustment for cancer stage.” Pages 9/10

“In FPS, the results were consistent with those observed in the UK Biobank. Psychological distress was associated with a 1.72 times increased risk of all-cause mortality after multivariable adjustment (95% CI: 1.44-2.07) (Table 3). Additional adjustment for cancer stage attenuated the association by 22%, although the relationship remained statistically significant (Table 4).” Page 12

“In UK Biobank, psychological distress was associated with a 1.45-fold increased risk of cancer mortality after adjustment for all covariates (95% CI, 1.21-1.74) (Table 5). Additional adjustment for cancer stage attenuated the association by 25%, although the relationship remained statistically significant (Table 6).” Page 12/13

“In the FPS, additional adjustment for cancer stage to account for cancer severity led to substantial attenuation of associations between psychological distress and all-cause or cancer mortality, although the relationships remained.” Page 16

Table 2. Characteristics of individuals diagnosed with cancer within 4 years prior to baseline assessments in the UK Biobank and Replication Cohort (Finnish Public Sector Study)

UK Biobank ^a			Finnish Public Sector Study		
Total	Died	Alive	Total	Died (N =	Alive (N
(N =	(N =	(N =	(N =	505)	= 5234)
13349)	2421)	10928)	5739)		

	Mean (SD)/n (%)	Mean (SD)/n (%)	Mean (SD)/n (%)	Mean (SD)/n (%)	Mean (SD)/n (%)	Mean (SD)/n (%)
Cancer stage	-	-	-			
Unknown				1512 (26.4)	64 (12.7)	1448 (27.7)
Localised				2779 (48.4)	149 (29.5)	2630 (50.3)
Non-localised, only regional lymph node metastases				821 (14.3)	113 (22.4)	708 (13.5)
Metasised farther than to regional lymph nodes or invades adjacent tissues				274 (4.8)	88 (17.4)	186 (3.6)
Non-localised, no information on extent				167 (2.9)	35 (6.9)	132 (2.5)
Locally advanced, tumour invades adjacent tissues				68 (1.2)	11 (2.2)	57 (1.1)
Non-localised, also distant lymph node metastases				118 (2.1)	45 (8.9)	73 (1.4)

Table 4. Hazard ratios for association between psychological distress and all-cause mortality after adjustment for each mediator.

	UK Biobank		FPS	
Model 2	Adjusted HRs (95% CI)	-	Adjusted HRs (95% CI)	-
	1.43 (1.23-1.67)	-	1.72 (1.44-2.07)	-
Model 2 + cancer stage	Adjusted HRs (95% CI)	-	Adjusted HRs (95% CI)	% of attenuation
	-	-	1.56 (1.29-1.88)	22.2
Model 2 + mediator	Adjusted HRs (95% CI)	% of attenuation	Adjusted HRs (95% CI)	% of attenuation
Inflammation	1.35 (1.15-1.57)	18.6	-	-
Fruit	1.42 (1.22-1.66)	2.3	-	-
Vegetables	1.43 (1.23-1.67)	0.0	-	-
Red meat	1.43 (1.23-1.67)	0.0	-	-
Processed meat	1.43 (1.23-1.67)	0.0	-	-
Physical activity	1.40 (1.20-1.63)	7.0	1.68 (1.40-2.02)	5.6
BMI	1.44 (1.23-1.67)	0.0	1.73 (1.44-2.08)	0.0
Alcohol	1.42 (1.22-1.66)	2.3	1.73 (1.44-2.08)	0.0
Smoking	1.36 (1.17-1.59)	16.3	1.71 (1.43-2.06)	1.4
Sleep quality	1.42 (1.21-1.65)	2.3	1.70 (1.40-2.06)	2.8
Sleep duration	1.39 (1.19-1.62)	9.3	1.75 (1.45-2.11)	0.0

All models adjusted for age, sex, ethnicity, education, number of comorbidities, age at cancer diagnosis, time between cancer diagnosis and depressive symptoms assessment, and antidepressant medication (yes/no).

Table 6. Hazard ratios for association between psychological distress and cancer mortality after adjustment for each mediator.

	UK Biobank		FPS	
Model 2	Adjusted HRs (95% CI)	-	Adjusted HRs (95% CI)	-
	1.45 (1.21-1.74)	-	1.72 (1.41-2.10)	-
Model 2 + cancer stage	Adjusted HRs (95% CI)	-	Adjusted HRs (95% CI)	% of attenuation
	-	-	1.54 (1.26-1.90)	25.0
Model 2 + mediator	Adjusted SHRs (95% CI)	% of attenuation	Adjusted SHRs (95% CI)	% of attenuation
Inflammation	1.37 (1.14-1.64)	17.8	-	-
Fruit	1.44 (1.20-1.72)	2.2	-	-
Vegetables	1.45 (1.21-1.74)	0.0	-	-
Red meat	1.45 (1.21-1.74)	0.0	-	-
Processed meat	1.45 (1.21-1.74)	0.0	-	-
Physical activity	1.42 (1.18-1.70)	6.7	1.68 (1.38-2.05)	5.6
BMI	1.45 (1.21-1.74)	0.0	1.72 (1.41-2.11)	0.0
Alcohol	1.44 (1.20-1.73)	2.2	1.73 (1.41-2.11)	0.0
Smoking	1.39 (1.16-1.67)	13.3	1.72 (1.40-2.10)	0.0
Sleep quality	1.44 (1.20-1.73)	2.2	1.66 (1.35-2.04)	8.3
Sleep duration	1.40 (1.20-1.69)	11.1	1.75 (1.43-2.14)	0.0

All models adjusted for age, sex, ethnicity, education, number of comorbidities, age at cancer diagnosis, time between cancer diagnosis and depressive symptoms assessment, and antidepressant medication (yes/no).

Supplementary table 14. Association between psychological distress and all-cause mortality risk, additionally adjusting for cancer stage (FPS).

All-cause mortality		
	HR (95% CI)	p value
N deaths/N total = 505/5739		
Model 1	1.80 (1.50-2.16)	<0.001***
Model 2	1.72 (1.44-2.07)	<0.001***
N deaths/N total = 326/5112		
Model 2 (excluded stages 3-6)	1.56 (1.24, 1.97)	0.0002**
N deaths/N total = 213/4291		
Model 2 (excluded stages 2-6)	1.71 (1.28, 2.28)	0.0002**

Notes. **p*<0.05; ***p*<0.01, ****p*<0.001.

CI = confidence intervals, HR = hazard ratio.

Model 1 adjusted for age and sex.

Model 2 adjusted for age, sex, ethnicity, education, number of comorbidities, age at cancer diagnosis, time between cancer diagnosis and depressive symptoms assessment, and antidepressant medication (yes/no).

Supplementary table 15. Association between psychological distress and cancer mortality risk, additionally adjusting for cancer stage (FPS).

	Cancer mortality	
	SHR (95% CI)	p value
N deaths/N total = 440/5739		
Model 1	1.70 (1.47, 2.18)	<0.001***
Model 2	1.72 (1.41, 2.10)	<0.001***
N deaths/N total = 269/5112		
Model 2 (excluded stages 3-6)	1.57 (1.21, 2.03)	0.0007**
N deaths/N total = 169/4291		
Model 2 (excluded stages 2-6)	1.76 (1.28, 2.43)	0.0006**

Notes. **p*<0.05; ***p*<0.01, ****p*<0.001.

CI = confidence intervals, HR = hazard ratio.

Model 1 adjusted for age and sex.

Model 2 adjusted for age, sex, ethnicity, education, number of comorbidities, age at cancer diagnosis, time between cancer diagnosis and depressive symptoms assessment, and antidepressant medication (yes/no).

2. In the mediation analysis, the authors assessed diet, physical activity, alcohol consumption, smoking, sleep duration and sleep quality. All these might be causes of distress, and in a cross-sectional assessment at the baseline, it is not sure about the causality, which is quite important for mediation analysis.

- We thank the reviewer for this insightful comment. We agree that a key limitation of our mediation analysis is the cross-sectional nature of the baseline data, which prevents us from establishing causal relationships between psychological distress and the mediators, which is crucial for understanding the direction of mediation effects. We have added this limitation in our Strengths and Limitations section on Page 18.

“Second, we did not have repeated measures on psychological distress, limiting our ability to assess how changes in distress over time affects survival. While we excluded participants who died within a year of baseline to address reverse causality, the possibility that undiagnosed cancer progression could cause both distress and mortality remains. Longitudinal assessments of distress would be beneficial to investigate how changes in distress over time might influence cancer progression and survival. Future work is warranted to investigate the association between trajectories of distress and survival to identify optimal timepoints for intervention.” Page 18

3. The underlying time scale for the analysis is not clear. It should be time since diagnosis for cancer survival.

- Thank you for this comment. We have conducted an additional sensitivity analysis using time since diagnosis as the underlying time scale. We have added the Methods to Pages 10 and 11 and the Results to Pages 14 and 15. The results are also presented in Supplementary Tables 8 and 9.
- The results were similar to the main analyses for both all-cause and cancer mortality. For all-cause mortality, after multivariable adjustment the HR was 1.43, compared to 1.43 in the main analysis using time since baseline assessments as the underlying time scale. For cancer mortality, after multivariable adjustment the HR was 1.44, compared to 1.45 in the main analysis using time since baseline assessments as the underlying time scale.

“Sensitivity analyses. Additional sensitivity analyses were carried out in our sample of UK Biobank participants (primary analysis). First, the analyses were run using a continuous measure of psychological distress rather than a binary measure. Second, participants were excluded if they died within one year of baseline assessments to account for reverse causality. Third, the analyses were repeated using self-reported comorbidities instead of hospital records. Self-reported comorbidities included diabetes, Alzheimer’s, asthma, dementia, Parkinson’s, hypertension, angina, heart attack, stroke, heart failure, heart murmur, abnormal heart rhythm, and chronic kidney disease. The total number of comorbidities was collapsed into 0/1+. Fourth, the analyses were repeated in a sample with no missing data on the exposure, covariates, mediators, and outcomes. **Fifth, the analyses were run using time since diagnosis (rather than time since baseline) as the underlying time scale.** Sixth, the analyses were run excluding individuals with non-melanoma skin cancer.” Pages 10 and 11

*“Sensitivity analyses conducted in UK Biobank yielded similar results to the main analyses (1) when distress was measured continuously rather than categorically (Supplementary Tables 2 and 3), (2) after excluding people who died within a year of baseline assessments to account for reverse causality (Supplementary Tables 4 and 5), (3) when self-reported comorbidities rather than comorbidities ascertained through hospital records was adjusted for (Supplementary Tables 6 and 7), (4) **when time from diagnosis was used as the underlying time scale** (Supplementary Tables 8 and 9), and (5) excluding individuals diagnosed with non-melanoma skin cancer (Supplementary Tables 10 and 11).” Page 14/15*

Supplementary table 8. Association between psychological distress and all-cause mortality risk over 19 years of follow-up among people LWBC, using time from diagnosis as the underlying time scale (UK Biobank).

	All-cause mortality	
	HR (95% CI)	p value
	N deaths /N total = 2421/13349	
Model 1	1.69 (1.46, 1.96)	<0.001***
Model 2	1.43 (1.23, 1.67)	<0.001***

Notes. *p<0.05; **p<0.01, ***p<0.001.

CI = confidence intervals, HR = hazard ratio.

Model 1 adjusted for age and sex.

Model 2 adjusted for age, sex, ethnicity, education, number of comorbidities, age at cancer diagnosis, time between cancer diagnosis and depressive symptoms assessment, and antidepressant medication (yes/no).

Supplementary table 9. Association between psychological distress and cancer-specific mortality risk over 19 years of follow-up among people LWBC, using time from diagnosis as the underlying time scale (UK Biobank).

	Cancer-specific mortality	
	SHR (95% CI)	p value
N deaths/N total = 1790/13349		
Model 1	1.59 (1.34, 1.89)	<0.001***
Model 2	1.44 (1.20, 1.73)	<0.001***

Notes. * $p < 0.05$; ** $p < 0.01$, *** $p < 0.001$.

CI = confidence intervals, SHR = sub-distribution hazard ratio.

Model 1 adjusted for age and sex.

Model 2 adjusted for age, sex, ethnicity, education, number of comorbidities, age at cancer diagnosis, time between cancer diagnosis and depressive symptoms assessment, and antidepressant medication (yes/no).

4. Taking a look at the details of different cancer types for the two cohorts, you will find that the results are not consistent between them in specific cancers.

- There are some differences in the prevalence of different cancer types in the two cohorts. For example, breast cancer is more prevalent in the FPS whereas non-melanoma skin cancer is much more prevalent in the UK Biobank cohort. These differences might reflect the differences in sex between the two cohorts – there is a greater proportion of females in the FPS compared to the UK Biobank, potentially explaining why breast cancer is much more common in the FPS. There are also some differences in the cancer type-stratified analysis results between the two cohorts. For example, in the analysis examining the association between distress and all-cause mortality, there is an association for people with non-melanoma skin cancer, breast cancer and multiple myeloma in UK Biobank but not FPS. These differences may be partially attributable to the differences in participant characteristics and cancer type distributions between the two samples. Also, the FPS is a more highly educated sample and report higher levels of distress.

5. Why Finnish Public Sector Study has so many females?

- The Finnish Public Sector study (FPS) sample is drawn from municipal employees in Finland, particularly in healthcare, education, and social services. These sectors are predominantly staffed by women in Finland.

In response to these two comments, we have added the following paragraph to the Discussion on Pages 17 and 18:

“Stratified analyses revealed that the associations between distress and survival varied by cancer type. However, there were some inconsistencies between the two cohorts in these results. For example, in the UK Biobank, distress was associated with increased risk for all-cause mortality among individuals with breast cancer, non-melanoma skin cancer, and multiple myeloma, whereas no such associations were observed in the FPS cohort. These differences may be partially attributable to differences in participant characteristics and cancer type distributions between the two cohorts. Although the absolute number of breast cancer cases was similar in both cohorts, breast cancer represented a greater proportion of cancer cases in the FPS sample (36.8% vs 19.6%). This likely reflects the higher proportion of women in FPS (82.8% vs 51.2%), as the sample is drawn from municipal employees in Finland, particularly in healthcare, education, and social services. These sectors are predominantly staffed by women. In contrast, non-melanoma skin cancer was more prevalent in the UK Biobank cohort (29.5% vs 12.5%). Furthermore, FPS participants were more highly educated and more likely to experience high levels of psychological distress. Additionally, a greater proportion of UK Biobank participants died before the end of follow-up compared to those in FPS (18.1% vs 8.8%), which may influence the observed patterns.” (Pages 17 and 18)

Reviewer #3:

This study investigates the association between psychological distress and mortality in people living with and beyond cancer (LWBC), exploring potential mediators such as inflammation and health behaviors. The research is well-designed, leveraging two large cohorts (UK Biobank and Finnish Public Sector Study), and addresses an important gap in psycho-oncology literature. The findings are robust, with consistent results across cohorts, and the manuscript is clearly written. However, some issues need clarification as follows.

1. The lack of data on cancer stage, severity, and treatment is a significant limitation. These factors could confound the observed associations between distress and mortality. For instance, advanced cancer stages or aggressive treatments might independently increase both distress and mortality risk. The authors should explicitly discuss this limitation and consider how it might affect the interpretation of their results. If possible, sensitivity analyses excluding participants with advanced-stage cancer could strengthen the findings.

- The UK Biobank has not released data on cancer stage; however, we have conducted two additional analyses in the Finnish Public Sector (FPS) study (1) including cancer stage as an additional covariate in the association between psychological distress and all-cause/cancer mortality and (2) excluding people with advanced stage cancer.
- These analyses found that cancer stage attenuates but does not remove the association between distress and all-cause mortality.
- We have added the Methods for these analyses on Pages 9/10 and Page 11 (sensitivity analysis), descriptive statistics for cancer stage in Table 2, the results on Pages 12/13 and 15 and in Tables 4 and 6, as well as Supplementary Tables 14 and 15, and a brief Discussion of the findings on Page 16. Unfortunately, neither cohort study has data on cancer treatments, and we acknowledge this as a limitation of our work in the ‘Strengths and Limitations’ section of the Discussion (Page 19).

“Effect estimates were adjusted for (1) age and sex (minimally adjusted model), (2) age, sex, sociodemographic factors, age at cancer diagnosis, time between cancer diagnosis and depressive symptoms assessment, and antidepressant use (multivariable adjusted model). In FPS, a third model was fitted with additional adjustment for cancer stage.” Pages 9/10

“In FPS, the results were consistent with those observed in the UK Biobank. Psychological distress was associated with a 1.72 times increased risk of all-cause mortality after multivariable adjustment (95% CI: 1.44-2.07) (Table 3). Additional adjustment for cancer stage attenuated the association by 22%, although the relationship remained statistically significant (Table 4).” Page 12

“In UK Biobank, psychological distress was associated with a 1.45-fold increased risk of cancer mortality after adjustment for all covariates (95% CI, 1.21-1.74) (Table 5). Additional adjustment for cancer stage attenuated the association by 25%, although the relationship remained statistically significant (Table 6).” Page 12/13

“In the FPS, additional adjustment for cancer stage to account for cancer severity led to substantial attenuation of associations between psychological distress and all-cause or cancer mortality, although the relationships remained.” Page 16

“First, we lacked data on cancer treatment. This meant we were unable to account for the potential impact of cancer treatments on associations between distress and survival.”

Additionally, we could not account for the potential impact of cancer treatments on the immune system when exploring inflammation as a potential mediator.” Page 19

Table 2. Characteristics of individuals diagnosed with cancer within 4 years prior to baseline assessments in the UK Biobank and Replication Cohort (Finnish Public Sector Study)

	UK Biobank ^a			Finnish Public Sector Study		
	Total (N = 13349)	Died (N = 2421)	Alive (N = 10928)	Total (N = 5739)	Died (N = 505)	Alive (N = 5234)
	Mean (SD)/n (%)	Mean (SD)/n (%)	Mean (SD)/n (%)	Mean (SD)/n (%)	Mean (SD)/n (%)	Mean (SD)/n (%)
Cancer stage	-	-	-			
Unknown				1512 (26.4)	64 (12.7)	1448 (27.7)
Localised				2779 (48.4)	149 (29.5)	2630 (50.3)
Non-localised, only regional lymph node metastases				821 (14.3)	113 (22.4)	708 (13.5)
Metasised farther than to regional lymph nodes or invades adjacent tissues				274 (4.8)	88 (17.4)	186 (3.6)
Non-localised, no information on extent				167 (2.9)	35 (6.9)	132 (2.5)
Locally advanced, tumour invades adjacent tissues				68 (1.2)	11 (2.2)	57 (1.1)
Non-localised, also distant lymph node metastases				118 (2.1)	45 (8.9)	73 (1.4)

Table 4. Hazard ratios for association between psychological distress and all-cause mortality after adjustment for each mediator.

	UK Biobank		FPS	
Model 2	Adjusted HRs (95% CI)	-	Adjusted HRs (95% CI)	-
	1.43 (1.23-1.67)	-	1.72 (1.44-2.07)	-
Model 2 + cancer stage	Adjusted HRs (95% CI)	-	Adjusted HRs (95% CI)	% of attenuation
	-	-	1.56 (1.29-1.88)	22.2
Model 2 + mediator	Adjusted HRs (95% CI)	% of attenuation	Adjusted HRs (95% CI)	% of attenuation
Inflammation	1.35 (1.15-1.57)	18.6	-	-
Fruit	1.42 (1.22-1.66)	2.3	-	-
Vegetables	1.43 (1.23-1.67)	0.0	-	-
Red meat	1.43 (1.23-1.67)	0.0	-	-
Processed meat	1.43 (1.23-1.67)	0.0	-	-
Physical activity	1.40 (1.20-1.63)	7.0	1.68 (1.40-2.02)	5.6
BMI	1.44 (1.23-1.67)	0.0	1.73 (1.44-2.08)	0.0
Alcohol	1.42 (1.22-1.66)	2.3	1.73 (1.44-2.08)	0.0
Smoking	1.36 (1.17-1.59)	16.3	1.71 (1.43-2.06)	1.4
Sleep quality	1.42 (1.21-1.65)	2.3	1.70 (1.40-2.06)	2.8
Sleep duration	1.39 (1.19-1.62)	9.3	1.75 (1.45-2.11)	0.0

All models adjusted for age, sex, ethnicity, education, number of comorbidities, age at cancer diagnosis, time between cancer diagnosis and depressive symptoms assessment, and antidepressant medication (yes/no).

Table 6. Hazard ratios for association between psychological distress and cancer mortality after adjustment for each mediator.

	UK Biobank		FPS	
Model 2	Adjusted HRs (95% CI)	-	Adjusted HRs (95% CI)	-
	1.45 (1.21-1.74)	-	1.72 (1.41-2.10)	-
Model 2 + cancer stage	Adjusted HRs (95% CI)	-	Adjusted HRs (95% CI)	% of attenuation
	-	-	1.54 (1.26-1.90)	25.0
Model 2 + mediator	Adjusted SHRs (95% CI)	% of attenuation	Adjusted SHRs (95% CI)	% of attenuation
Inflammation	1.37 (1.14-1.64)	17.8	-	-
Fruit	1.44 (1.20-1.72)	2.2	-	-
Vegetables	1.45 (1.21-1.74)	0.0	-	-
Red meat	1.45 (1.21-1.74)	0.0	-	-
Processed meat	1.45 (1.21-1.74)	0.0	-	-
Physical activity	1.42 (1.18-1.70)	6.7	1.68 (1.38-2.05)	5.6
BMI	1.45 (1.21-1.74)	0.0	1.72 (1.41-2.11)	0.0
Alcohol	1.44 (1.20-1.73)	2.2	1.73 (1.41-2.11)	0.0
Smoking	1.39 (1.16-1.67)	13.3	1.72 (1.40-2.10)	0.0
Sleep quality	1.44 (1.20-1.73)	2.2	1.66 (1.35-2.04)	8.3
Sleep duration	1.40 (1.20-1.69)	11.1	1.75 (1.43-2.14)	0.0

All models adjusted for age, sex, ethnicity, education, number of comorbidities, age at cancer diagnosis, time between cancer diagnosis and depressive symptoms assessment, and antidepressant medication (yes/no).

Supplementary table 14. Association between psychological distress and all-cause mortality risk, additionally adjusting for cancer stage (FPS).

All-cause mortality		
	HR (95% CI)	p value
N deaths/N total = 505/5739		
Model 1	1.80 (1.50-2.16)	<0.001***
Model 2	1.72 (1.44-2.07)	<0.001***
N deaths/N total = 326/5112		
Model 2 (excluded stages 3-6)	1.56 (1.24, 1.97)	0.0002**
N deaths/N total = 213/4291		
Model 2 (excluded stages 2-6)	1.71 (1.28, 2.28)	0.0002**

Notes. **p*<0.05; ***p*<0.01, ****p*<0.001.

CI = confidence intervals, HR = hazard ratio.

Model 1 adjusted for age and sex.

Model 2 adjusted for age, sex, ethnicity, education, number of comorbidities, age at cancer diagnosis, time between cancer diagnosis and depressive symptoms assessment, and antidepressant medication (yes/no).

Supplementary table 15. Association between psychological distress and cancer mortality risk, additionally adjusting for cancer stage (FPS).

	Cancer mortality	
	SHR (95% CI)	p value
N deaths/N total = 440/5739		
Model 1	1.70 (1.47, 2.18)	<0.001***
Model 2	1.72 (1.41, 2.10)	<0.001***
N deaths/N total = 269/5112		
Model 2 (excluded stages 3-6)	1.57 (1.21, 2.03)	0.0007**
N deaths/N total = 169/4291		
Model 2 (excluded stages 2-6)	1.76 (1.28, 2.43)	0.0006**

Notes. * $p < 0.05$; ** $p < 0.01$, *** $p < 0.001$.

CI = confidence intervals, HR = hazard ratio.

Model 1 adjusted for age and sex.

Model 2 adjusted for age, sex, ethnicity, education, number of comorbidities, age at cancer diagnosis, time between cancer diagnosis and depressive symptoms assessment, and antidepressant medication (yes/no).

2. The use of non-cancer-specific scales (PHQ-4 and GHQ-12) may not fully capture the unique distress experienced by cancer patients. The authors should discuss whether these tools are validated in cancer populations and how this might influence the results. Additionally, the binary categorization of distress (high vs. low) may oversimplify the relationship. The sensitivity analysis using continuous distress scores is helpful, but further discussion on the implications of this choice is needed.

- Thank you for your valuable comments.
- The PHQ-4 and GHQ-12 have both been validated in people diagnosed with cancer, and we have now moved this validation information from the Supplementary File to the Methods section of the manuscript (Page 7).

“In UK Biobank, psychological distress was measured at baseline using the 4-item version of the Patient Health Questionnaire (PHQ-4) (Supplementary file).²⁵ The established cut-off score of 6 or more was used to denote psychological distress.^{26,27} The PHQ-4 has been shown to be a valid measure of psychological distress in people diagnosed with cancer.^{28,29}” Page 7

“In FPS, distress was measured at baseline using the 12-item General Health Questionnaire (GHQ-12).³⁰ Based on the optimal cut-off point of 3 established in a Finnish validation study,³¹ participants were categorised as non-distressed (GHQ scores 0-3) and distressed (GHQ scores 4-12). The GHQ-12 has been shown to be a valid measure of psychological distress in people LWBC.^{32,33}” Page 7

- We also acknowledge in the Strengths and Limitations section of the discussion that, while these tools are commonly used for assessing distress in a range of populations, they may not fully capture the unique distress experienced by people diagnosed with cancer (Page 18).

“Fifth, psychological distress was assessed using non cancer-specific scales, which may not fully capture the unique distress experienced by people LWBC.” Page 18

- We also appreciate the reviewer’s point about the binary categorisation of distress potentially oversimplifying the relationship between distress and survival. In response, we have further discussed the implications of this choice in the Discussion (Page 15), highlighting the potential limitations and providing additional context regarding the continuous distress scores using in sensitivity analyses.

“The observed hazard ratios of 1.54 and 1.57 for psychological distress in relation to all-cause mortality and cancer mortality in people LWBC are higher than those reported in previous meta-analyses (range: 1.24-1.40 for all-cause mortality and 1.21-1.39 for cancer mortality).^{12,38,39} These differences may be partially attributed to our use of a binary categorisation of distress, which could oversimplify the relationship between distress and mortality. To address this, we also conducted sensitivity analyses using continuous distress scores, which showed similar associations, suggesting that the binary classification did not unduly influence the results.” Page 15

3. The study focuses solely on C-reactive protein (CRP) as an inflammatory marker. Given the complexity of inflammation, incorporating additional biomarkers (e.g., IL-6, TNF- α) would provide a more comprehensive understanding. The authors should acknowledge this limitation and suggest it as a direction for future research.

- We agree that focusing solely on CRP as an inflammatory marker may limit the scope of our analysis, given the complexity of the inflammatory process. We have acknowledged this limitation in the ‘Strengths and Limitations’ section of the Discussion (Page 18), and we highlight the need for future research to incorporate additional biomarkers, such as IL-6 and TNF- α . Including these markers would provide a more comprehensive understanding of the inflammatory mechanisms involved in the relationship between psychological distress and mortality in people living with and beyond cancer.

“Fourth, our focus on a single inflammatory marker, C-reactive protein, may limit the interpretation of the findings, as inflammation is a complex, systemic process. Future work could incorporate additional systemic biomarkers of inflammation, such as tumour necrosis factor alpha (TNF- α) or interleukin-6 (IL-6), to provide a more comprehensive understanding of inflammatory pathways.” Page 19

4. While the authors conducted sensitivity analyses excluding deaths within one year of baseline, the possibility of reverse causality(e.g., undiagnosed cancer progression causing distress) remains. The discussion should elaborate on this and consider whether longitudinal assessments of distress could mitigate this issue.

- Thank you for this insightful comment. We acknowledge that the possibility of reverse causality still remains, as undiagnosed cancer progression could lead to both increased distress and mortality. To address this, we have elaborated on this issue, emphasising that longitudinal assessments of distress would help mitigate this concern by providing insight into how changes in distress over time may influence both cancer progression and survival (Page 17 and 18).

“Second, we did not have repeated measures on psychological distress, limiting our ability to assess how changes in distress over time affects survival. While we excluded participants who died within a year of baseline to address reverse causality, the possibility that undiagnosed cancer progression could cause both distress and mortality remains. Longitudinal assessments of distress would be beneficial to investigate how changes in distress over time might influence cancer progression and survival. Future work is warranted to investigate the association between trajectories of distress and survival to identify optimal timepoints for intervention.” Pages 17 and 18

5. The stratified analyses by cancer type are insightful but raise questions about the mechanisms underlying the observed variations. For example, why is distress associated with higher mortality in head and neck cancer but not in gastrointestinal cancer? The authors should hypothesize about these differences, possibly linking them to cancer-specific pathways or treatment effects.

- We agree that the variations observed in the stratified analyses by cancer type raise important questions. In response, we have expanded our discussion to hypothesize about potential reasons underlying these differences. Specifically, we note that individuals with head and neck cancers may be particularly vulnerable to psychological distress due to the emotional and physical toll of the cancer itself and treatments (causing difficulties with swallowing, speech, breathing, and appearance). In contrast, cancers with more predictable progression and standardised treatment, such as testicular cancer, may be less impacted by psychological factors. We have included this into the Discussion section (Page 18).

“Other meta-analyses have also found that the association between distress and mortality in people LWBC may differ by cancer type.^{12,38,52} Consistent with our results, associations between distress and poorer survival have been demonstrated in head and neck,⁵³ breast,³⁸ and colorectal cancer.⁵² Research suggests that psychological factors may have a smaller impact on cancer types that typically follow a predictable pattern of progression.¹¹ Individuals with head and neck cancers may be particularly vulnerable to psychological distress due to the disease itself causing difficulty with swallowing, breathing, and speaking, as well as treatment side effects such as dry mouth and changes in appearance.⁵⁴ These factors may amplify the influence of distress on survival. In contrast, cancers with more stable trajectories, such as testicular cancer, may be less influenced by psychological factors, due to the relative predictability of treatment and disease progression. Together, these findings suggest that the mechanisms linking psychological distress to cancer outcomes may vary by cancer type. Future research should focus on further exploring the association between distress and mortality, as well as the underlying mechanisms, in specific cancer types.” Page 18

6. The use of multiple imputation for missing data is appropriate, but the authors should provide more details on the variables included in the imputation model and the proportion of missing data for key variables.

- Thank you for this comment. We have added details on the variables included in the imputation model, which were all variables used in the main analyses (exposure, covariates, mediators, and outcomes), as well as auxiliary variables likely related to the missingness (income). This is detailed on Page 9 in the ‘Statistical analysis’ section. We also provide the proportion of missing data for key variables in Supplementary Table 1. We have added a sentence in the ‘Statistical analysis’ section of the manuscript about the range of missing data for key variables, which was from 0% (age, sex, comorbidities) to 24% (physical activity) (Page 9).

“In UK Biobank, missing data on predictors and covariates were assumed to be missing at random and imputed using multiple imputation by chained equations.³⁵ Twenty datasets were imputed, and estimates were pooled using Rubin’s rules (Rubin, 2004). The imputation model included all variables used in the main analyses (exposure, covariates, mediators, and outcomes), as well as auxiliary variables likely related to the missingness (income). The proportion of missing data for key variables ranged from 0% (e.g., age, sex, number of comorbidities) to approximately 24% (e.g., physical activity).” Page 9

7. The methods for assessing mediators (e.g., diet, physical activity) vary between cohorts and rely largely on self-report, which may introduce bias. The authors should briefly discuss the validity of these measures and any potential misclassification.

- We acknowledge that it is a limitation that the methods for assessing mediators vary between cohorts and rely largely on self-report, which may introduce bias due to recall and social desirability biases. This could have affected the validity of the results. However, research has shown that the tools to measure diet and physical activity in the UK Biobank are valid and reliable. Nonetheless, future work could still benefit from incorporating objective measures of physical activity, such as accelerometers, and biomarkers for diet to reduce potential bias and improve the validity of the findings. We have added this limitation and suggestion for future work in the ‘Strengths and Limitations’ section of the discussion on Pages 19 and 20.

“Sixth, the methods for assessing mediators varied between cohorts and relied largely on self-report. This reliance on self-report may introduce bias due to recall and social desirability biases, affecting the accuracy of the results. However, previous research shows that the self-report diet questions in the UK Biobank reliably rank participants according to intakes of the main food groups.⁵⁵ Similarly, the International Physical Activity Questionnaire (IPAQ), used to assess physical activity in the UK Biobank, has been shown to be a valid tool.⁵⁶ Nonetheless, future research could benefit from incorporating objective measures of physical activity, such as accelerometers, and biomarkers for diet to reduce potential bias and improve the validity of the findings.” Page 19 and 20

- We also addressed any potential misclassification of the mediators by repeating our mediation analyses in two different cohorts using the exact same classifications. The results for the mediators for each cohort were similar.

8. The formula for calculating attenuation percentages is clear, but the authors should confirm whether this method is appropriate for Cox models, as other approaches (e.g., mediation analysis with bootstrapping) might provide more robust estimates.

- Thank you for this comment. Attenuation of excess risk of an appropriate way to assess mediation in Cox models and has been used in prior research. We chose this approach due to the exploratory nature of our study and its straightforward application.
- We thank the reviewer for their insightful suggestion to use alternative mediation analysis tools to assess mediation for more robust estimates.
- To address this, we have conducted an additional mediation analysis using the inverse odds ratio weighted method. This approach allows simultaneous consideration of multiple mediators to decompose the total effect into direct and indirect methods.
- We have included the methods for these additional analyses (Page 10), the results (Page 14), and an interpretation (Page 18) in the revised manuscript, which strengthens the robustness of our findings by methodological triangulation (Page 20)

“To account for mediation involving multiple interrelated mediators simultaneously, we also applied an inverse odds ratio-weighted method specifically for the outcome of all-cause mortality³⁸. This approach decomposes the total association between psychological distress and mortality (total effect) into a natural direct effect (NDE), not operating through the mediators, and a natural indirect effect (NIE), operating through the mediators jointly. In FPS, this analysis was conducted using both the multivariable adjusted model (Model 2) and a further model adjusting for cancer stage (Model 3).” Page 10

“To account for mediation by multiple biological and behavioural factors simultaneously, we applied the inverse odds ratio-weighted method. In UK Biobank, this approach indicated that approximately 29% (95% CI: -46.9, 104.8) of the association between psychological distress and all-cause mortality was mediated by the combined set of mediators. In FPS, the combined mediators accounted for 25.7% (95% CI: 1.7-49.7) of the association in the multivariable adjusted model (Model 2). After further adjustment for cancer stage (Model 3), the mediation estimate was similar (25.6%) but was no longer statistically significant (95% CI: -6.4, 57.5).” Page 14

“The mediators found in this study only contributed a moderate amount to the association between distress and mortality, suggesting that other factors, such as treatment adherence, may play a role. Research shows that depression predicts non-adherence to medication in people with cancer^{46,47} due to factors such as lack of motivation, decreased energy, or feelings of hopelessness.⁴⁸ Furthermore, non-adherence to medication has been associated with greater risk of mortality in people with cancer.⁴⁹⁻⁵¹ Additionally, distress might affect other aspects of treatment, such as attendance at follow-up appointments or participation in clinical trials, which might also affect survival outcomes. Another potential mediator is tumour response to treatment. Psychological distress can lead to inflammation, which may affect how tumours respond to treatment, ultimately influencing cancer progression and survival.⁵² Future research is needed to further explore the mechanisms through which psychological distress affects survival among people LWBC.” Page 18

“We also applied two mediation approaches, the percentage of attenuation method and the inverse odds ratio-weighted method, allowing for methodological triangulation that enhances the robustness of our findings.” Page 20

9. Table 2 is comprehensive but could be simplified for readability (e.g., collapsing some categories or moving detailed descriptions to supplementary materials)

- Thank you for your suggestion to simplify Table 2. While we understand the need for readability, we believe that collapsing certain categories or removing detailed descriptions could result in the loss of important information. The categories and detailed descriptions included in the table allow for a comprehensive comparison of key variables across the 2 cohorts and ensure transparency in reporting.

10. The abstract could be more concise; the methods section is overly detailed for an abstract.

- Thank you for this suggestion; we have made the Abstract more concise. See below for an updated Abstract.

“The biological and behavioural mechanisms linking psychological distress to excess mortality in people living with and beyond cancer (LWBC) remain poorly understood. In this multi-cohort study, we examined the association between psychological distress and both cancer-specific and all-cause mortality in individuals LWBC, and assessed the potential mediating roles of inflammation and health behaviours. The primary analysis included 13,349 adults from the UK Biobank (2006-2021), with replication in 5,739 participants from the Finnish Public Sector study (2000-2018). Psychological distress was associated with an increased risk of all-cause mortality (pooled adjusted risk ratio (RR), 1.54, 95% confidence interval (CI), 1.37-1.74) and cancer mortality (pooled RR, 1.57; 95% CI, 1.37-1.79). Systemic inflammation explained up to 18.6% of these associations, whereas diet, alcohol consumption, and body mass index did not mediate these relationships. Evidence for mediation by other health behaviours was inconsistent. While adjustment for cancer stage attenuated the distress-mortality link by up to 25%, psychological distress remained a robust predictor of both all-cause and cancer mortality. These findings suggest that psychological distress is an independent predictor of mortality risk in people LWBC, partially attributable to elevated levels of systemic inflammation.”

11. Some acronyms (e.g., LWBC) are defined multiple times; ensure consistency.

- Thank you for the comment. We have reviewed the manuscript and have ensured that all acronyms (e.g., LWBC, ICD) are only explained once.

REVIEWERS' COMMENTS

Reviewer #1 (Remarks to the Author):

I have no more questions.

- Thank you again to Reviewer 1 for all their helpful comments.

Reviewer #2 (Remarks to the Author):

The authors have answered all my concerns. One additional question is that mediation analysis was performed for all cause mortality. Why the authors did not perform it for cancer specific mortality?

- Initially we did not run the counterfactual mediation analysis for cancer mortality because the confidence intervals were already wide in the analyses of all-cause mortality; less cases would result in similarly wide or even wider confidence intervals.
- However, we have run and reported the counterfactual mediation analyses for cancer mortality in both the UK Biobank and Finnish Public Sector study. The results were very similar to the analysis for all-cause mortality, showing that the combined set of mediators accounted for approximately 30% of the association between distress and cancer mortality.

“To examine the combined mediating effect of biological and behavioural factors simultaneously, we applied the inverse odds ratio-weighted method. In UK Biobank, this approach indicated that approximately 30% (95% CI: -55.7, 115.7) of the association between psychological distress and cancer mortality was mediated by the combined set of mediators. In FPS, the combined mediators accounted for 28.5% (95% CI: 3.1-53.9) of the association in the multivariable adjusted model (Model 2). After further adjustment for cancer stage (Model 3), the mediation estimate was similar (30%) but was no longer statistically significant (95% CI: -6.5, 66.4).” (Page 9; Results section)

Reviewer #3 (Remarks to the Author):

The author has addressed all the issues raised in my review comments, and I have no further comments.

- Thank you again to Reviewer 3 for all their helpful comments.

Review Comments

General Assessment:

This study investigates the association between psychological distress and mortality in people living with and beyond cancer (LWBC), exploring potential mediators such as inflammation and health behaviors. The research is well-designed, leveraging two large cohorts (UK Biobank and Finnish Public Sector Study), and addresses an important gap in psycho-oncology literature. The findings are robust, with consistent results across cohorts, and the manuscript is clearly written. However, some issues need clarification as follows.

1. The lack of data on cancer stage, severity, and treatment is a significant limitation. These factors could confound the observed associations between distress and mortality. For instance, advanced cancer stages or aggressive treatments might independently increase both distress and mortality risk. The authors should explicitly discuss this limitation and consider how it might affect the interpretation of their results. If possible, sensitivity analyses excluding participants with advanced-stage cancer could strengthen the findings.
2. The use of non-cancer-specific scales (PHQ-4 and GHQ-12) may not fully capture the unique distress experienced by cancer patients. The authors should discuss whether these tools are validated in cancer populations and how this might influence the results. Additionally, the binary categorization of distress (high vs. low) may oversimplify the relationship. The sensitivity analysis using continuous distress scores is helpful, but further discussion on the implications of this choice is needed.
3. The study focuses solely on C-reactive protein (CRP) as an inflammatory marker. Given the complexity of inflammation, incorporating additional biomarkers (e.g., IL-6, TNF- α) would provide a more comprehensive understanding. The authors should acknowledge this limitation and suggest it as a direction for future research.
4. While the authors conducted sensitivity analyses excluding deaths within one year of baseline, the possibility of reverse causality

(e.g., undiagnosed cancer progression causing distress) remains. The discussion should elaborate on this and consider whether longitudinal assessments of distress could mitigate this issue.

5. The stratified analyses by cancer type are insightful but raise questions about the mechanisms underlying the observed variations. For example, why is distress associated with higher mortality in head and neck cancer but not in gastrointestinal cancer? The authors should hypothesize about these differences, possibly linking them to cancer-specific pathways or treatment effects.
6. The use of multiple imputation for missing data is appropriate, but the authors should provide more details on the variables included in the imputation model and the proportion of missing data for key variables.
7. The methods for assessing mediators (e.g., diet, physical activity) vary between cohorts and rely largely on self-report, which may introduce bias. The authors should briefly discuss the validity of these measures and any potential misclassification.
8. The formula for calculating attenuation percentages is clear, but the authors should confirm whether this method is appropriate for Cox models, as other approaches (e.g., mediation analysis with bootstrapping) might provide more robust estimates.
9. Table 2 is comprehensive but could be simplified for readability (e.g., collapsing some categories or moving detailed descriptions to supplementary materials)
10. The abstract could be more concise; the methods section is overly detailed for an abstract.
11. Some acronyms (e.g., LWBC) are defined multiple times; ensure consistency.

Conclusion:

This study makes a valuable contribution to understanding the link between psychological distress and mortality in cancer survivors, highlighting the roles of inflammation and health behaviors. Addressing the major concerns, particularly regarding cancer stage and treatment data, would strengthen

the manuscript.

Recommendation:

Major Revision